# Predicting the replicability of social and behavioural science claims in COVID-19 preprints

**A list of authors and their affiliations appears at the end of the paper**

Replications are important for assessing the reliability of published findings. However, they are costly, and it is infeasible to replicate everything. Accurate, fast, lower-cost alternatives such as eliciting predictions could accelerate assessment for rapid policy implementation in a crisis and help guide a more efficient allocation of scarce replication resources. We elicited judgements from participants on 100 claims from preprints about an emerging area of research (COVID-19 pandemic) using an interactive structured elicitation protocol, and we conducted 29 new high-powered replications. After interacting with their peers, participant groups with lower task expertise ('beginners') updated their estimates and confidence in their judgements significantly more than groups with greater task expertise ('experienced'). For experienced individuals, the average accuracy was 0.57 (95% CI: [0.53, 0.61]) after interaction, and they correctly classified 61% of claims; beginners' average accuracy was 0.58 (95% CI: [0.54, 0.62]), correctly classifying 69% of claims. The difference in accuracy between groups was not statistically significant and their judgements on the full set of claims were correlated ($r(98) = 0.48$, $P < 0.001$). These results suggest that both beginners and more-experienced participants using a structured process have some ability to make better-than-chance predictions about the reliability of 'fast science' under conditions of high uncertainty. However, given the importance of such assessments for making evidence-based critical decisions in a crisis, more research is required to understand who the right experts in forecasting replicability are and how their judgements ought to be elicited.

Over the past decade, several large-scale replication studies have called into question the reliability of published findings across disciplines including psychology, economics, computational social science, cancer biology and medicine[1–9]. Low replication rates may indicate credibility challenges for the published literature. More generally, they may undermine the trust of stakeholders and end-users who expect reliable evidence to guide practice and policymaking.

In the wake of this evidence, there is an increased appreciation for the role of replication across disciplines[10–14] and a slowly growing number of academic journals are encouraging the submission of replication studies[15–17].

However, replication studies are costly, time-consuming and sometimes logistically infeasible. It is impossible to replicate everything, and it is potentially a misuse of limited resources to try. Arguably, the return on replication investment would be greatest when the reliability of a given finding is uncertain and it has high potential of actual impact on the direction of research, policy or practical application[18,19]. Replications can then aid in confirming the initial results or identify

✉e-mail: alexandru.marcoci@gmail.com

boundary conditions on their generalizability and applicability. On the other hand, the expected utility of replication will be minimal when a published finding is seemingly incontrovertible, clearly generalizable and based on highly reliable methods, or conversely, extremely unlikely to be true or based on obsolete or debunked methods. To distinguish between what is worth replicating and what is not, we need methods to make accurate predictions about the credibility of research findings.

If predicting replication outcomes could be done quickly, with little cost and accurately, then those predictions could guide the allocation of limited resources for replication studies[19]. For example, accurate predictions could identify extremely likely and unlikely findings that do not require replication for credibility assessment and direct attention towards findings that are important but elicit high prediction uncertainty so that their replicability can be empirically verified. Moreover, predictions could identify heterogeneity in assumptions about the boundary conditions or necessary or sufficient features of the setting, methods and measures to observe the finding. This could guide the allocation of resources for replication or other credibility assessments to the areas of maximum ambiguity and importance. Accurate predictions about the replicability of research could also strengthen existing and overburdened scientific quality control mechanisms such as peer review, hiring and promotion decisions, and funding decisions by providing an additional indicator of credibility to guide further assessment[20].

Several studies have demonstrated that groups of experts have the potential to accurately predict the replicability of findings in the social and behavioural sciences[6,7,21,22]. Across these four studies, 'expert beliefs' on 103 published results were elicited using surveys and prediction markets. In contrast to the surveys, prediction markets generate a collective forecast through the trading of contracts (or 'bets') with payments that depend on the observed replication outcome. When interpreting elicited probabilities of less than 0.5 as forecasts for failed replications and elicited probabilities of more than 0.5 as forecasts for successful replications, markets achieved 73% accuracy. Moreover, prediction market prices correlated with replication outcomes ($r = 0.58$, a moderately strong effect[23]). Using similar methods, however, ref. 24 elicited expert predictions about 22 tests from DARPA's Next Generation Social Science programme[25] and found that both surveys and prediction markets achieved only 50% accuracy (Table 2 in ref. 24). The tests in this study were not direct replications but rather tested whether established theory applied to a novel context. A study on replications of 5 mouse model studies drawn from The Reproducibility Project: Cancer Biology (none of which replicated) suggests even poorer expert performance[26]. Using a survey of subjective probabilities (0–100%) of replication, ref. 26 found that experts correctly predicted whether a replication would find a statistically significant effect in the same direction as the original in only 30% of cases and whether a replication will have a similar or larger effect size in only 59% of cases.

Although they varied in their recruitment methods, all the above studies relied on experts for their replication judgements. Expertise was typically equated with having (or at least studying towards) a PhD in a cognate field. For instance, ref. 6 recruited from the Economic Science Association mailing list and the members of the Editorial Board of several top Economics journals, including American Economic Review, Quarterly Journal of Economics and Econometrica. The participants had to have at least a master's degree, and many were graduate students. Reference 21 recruited traders from the Open Science Framework and the Reproducibility Project: Psychology collaboration email lists who were active researchers in psychology, ranging from graduate students to professors. Reference 26 recruited experts on the basis of authorship of similar papers as those whose replications they were asked to predict. This particular sample was highly qualified, with an average of 89.5 (s.d. = 109.0) authored papers and 4,546 (s.d. = 7,932) citations.

In a recent study, however, ref. 27 elicited predictions on 27 studies extracted from the Many Labs 2 (ref. 22) and SSRP[7] replication projects and recruited 'laypeople' from MTurk and social media platforms, and first-year psychology students at the University of Amsterdam. The 233 participants provided judgements on replicability (binary measure: yes/no) across two conditions: study description-only ($n = 123$) and description-plus-evidence ($n = 110$). All participants read a short description of the study ('description-only'), and about half were also provided with information about Bayes Factors and their interpretation ('description-plus-evidence', for example, 'BF = 4.6, which qualifies as moderate evidence'). Reference 27 found that judgements in the description-only condition achieved 59% accuracy but improved to 67% in the description-plus-evidence condition. These findings raise interesting questions about the importance of expertise in the traditional sense (for example, peer esteem, relevant degrees) and how predictions are elicited.

Finally, participants in all the above studies evaluated the replicability of published research claims from top journals and established fields and traditions of enquiry. But not all scientific outputs fit this mould. Fast science is the "application-driven research confronted with an urgent need to accept or reject a certain hypothesis for the purposes of policy guidance, aimed at addressing a significant pending social harm" (ref. 28, p. 938). Social and behavioural science about the COVID-19 pandemic, for instance, is a prototypical example of fast science[29], and preprints have been an important vehicle through which fast COVID-19 science has been communicated for immediate policy decisions[30–32]. While assessing the reliability of preprints about a fast-developing public health crisis will share many of the challenges of evaluating peer-reviewed research, it will nevertheless entail much higher levels of uncertainty. In these cases, both individuals and organizations will have to make decisions about the quality of the scientific advice without help from domain experts. Indeed, during the COVID-19 pandemic, preprints informed experts, policymakers with little domain expertise, journalists and the general public. Given the new ways science is communicated (especially in a crisis), investigating different groups' ability to demarcate between reliable and unreliable claims under extreme uncertainty is even more critical.

The present study adds to the literature in four ways. First, our forecasting questions were drawn from research on the COVID-19 pandemic, using this as an example of an exceptionally rapidly evolving evidence base on which important policy and practice decisions were based. Second, we elicited predictions about the replicability of claims published in preprints rather than journal articles that have already been subject to rigorous peer review, thus introducing additional uncertainty. Third, we directly compared the performance of participant groups with lower task expertise ('beginners') to more-experienced groups. Neither beginners nor experienced participants had domain-relevant expertise in all the areas covered by our forecasting questions. Still, they had varying degrees of expertise in social and behavioural science methodologies and in evaluating quantitative social science research. Finally, we directly compared judgements made using a cooperative, interactive Delphi-style structured process with two elicitation rounds (ref. 33; Fig. 1) against the outcomes of a prediction market. With this study, we also release a new corpus of 35 replications, out of which we report 29 high-powered ones as the COVID-19 Preprint Replication Project (Table 1).

## Results

### The COVID-19 Preprint Replication Project

The COVID-19 Preprint Replication Project dataset was generated as part of the Defense Advanced Research Projects Agency (DARPA)'s Systematizing Confidence in Open Research and Evidence (SCORE) programme[34]. SCORE replications were conducted by a set of independent researchers worldwide with coordination provided by an organizing team based in the United States. Researchers self-selected

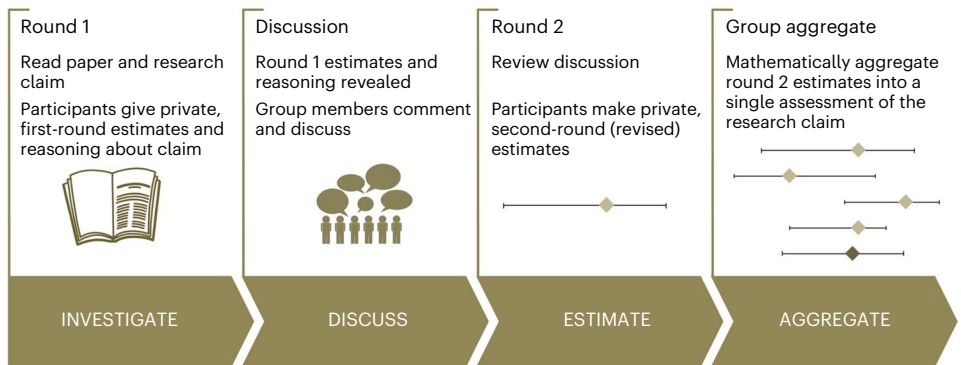

**Fig. 1 | The IDEA protocol.** The IDEA protocol as implemented on the repliCATS platform[33].

to replicate the COVID-19 claims on the basis of availability and expertise. Of the 100 eligible claims, 35 were replicated (Table 1), of which 29 were retained for further analysis. Of the proposed set of replication studies, 6 were considered underpowered. Results for all 35 replication studies are provided as Supplementary Information 2. Replications were defined as tests of the original claim using new data and an inferential test expected to be theoretically equivalent to what was reported in the preprint[35]. The process of replications was conducted following the same methodology used in the rest of the SCORE programme and resulted in two 'types' depending on the source of the new data.

New data replications focused on collecting new data as part of the replication attempt using data collection methods (for example, surveys), sampling techniques and an analytical approach as similar as possible to the original claim. Research teams that were identified as a match for conducting a replication of one of the eligible claims were responsible for obtaining ethics approval to conduct the study, documenting their process in a preregistration that underwent review before collecting the data and performing the analysis, and documenting the results in an outcome report.

Secondary data replications focused on identifying another existing data source that was not the same as the one used in the original study but was theoretically expected to test the original claim, usually by similar data collection methods, measures and sampling techniques. Secondary data replications were almost always split between two independent research teams. A data finder team was responsible for identifying the secondary data, preparing it into an analytic dataset and documenting their process in a preregistration and associated files. These data were passed to a data analysis team that would use the prepared dataset to complete the analysis portion of the preregistration, address reviewers' comments about the approach following a structured peer review process, analyse the data as prespecified following approval of the preregistration, and then document the results in an outcome report. Splitting the research into two teams preserved the independence of the analyst's plans since their decisions would not be overly influenced by familiarity with the dataset after selecting, cleaning and documenting it.

### Participant demographics

As anticipated, participants with lower task expertise ($N = 88$) differed significantly from the more-experienced participants ($N = 98$) on several relevant dimensions, including age, education, publication history, previous experience in metascience and statistical knowledge (Table 2). Note that we construe expertise in relation to the task of evaluating the quality of research in the social sciences and not expertise in the subject matter of the papers we assessed, which varied in terms of both discipline and methodological approach. Therefore, experienced participants were older, had attained higher levels of education, training and relevant experience, and performed significantly better on the quiz that tested their knowledge of statistical concepts and metaresearch. One participant

in the experienced sample who had only attained a high school degree when data collection started was excluded from all subsequent analyses.

### Replication outcomes and participants' predictions

All participants provided estimates in two elicitation rounds using an interval answer format. That is, participants were prompted to provide their (1) lower bound, (2) upper bound and (3) their best estimate of the probability a claim will replicate successfully (see Figs. 1 and 2 and Methods).

Of the 29 included replications, 16 were based on new data collection by the replication team ('new data replications'), and 13 were replications based on a secondary data source different from the original data ('secondary data replications'; see Methods). Among the former, 11 were successfully replicated (69%), while among the latter, 8 were successfully replicated (62%), for an overall replication rate of 65.4%. Experienced participants predicted a replication rate of 64.7% in Round 1 and 64.8% in Round 2. Beginners predicted a replication rate of 66.8% in Round 1 and a slightly higher rate of 69.1% in Round 2 (Fig. 3).

### Accuracy on claims with known replication outcomes

We used two accuracy measures in this study: error-based and classification accuracy. We calculated error-based accuracy as 1 minus the absolute difference between an estimate (0–1) and the replication outcome (0 for failed replications, 1 for successful replications). Classification accuracy was calculated as the proportion of estimates on the correct side of 0.5 (estimate ≥0.5 for successful replications, estimate <0.5 for failed replications) for a paper's replication outcome. Error-based accuracy was calculated at the individual and group level (that is, based on the pooled estimates within a given group).

The judgements of both beginners and more-experienced participants are consistent with some predictive ability, with both groups performing better than chance (that is, their 95% confidence intervals do not overlap 0.5) on most accuracy measures (Fig. 4). However, participants' predictions are only weakly correlated with the outcome of the replication ($r(27) = 0.180$, $P = 0.357$ for beginners and $r(27) = 0.251$, $P = 0.189$ for experienced participants). All models used are described in Table 3.

Wherever we found no differences between beginners and experienced participants, we further examined our non-significant results using two one-sided tests[36] at an alpha level of 0.05 and an equivalence bound of ±0.05 raw mean difference scores.

**Differences between beginners and experienced participants.** Against our preregistered hypothesis, we observed no significant difference between the error-based accuracy of beginners' and more-experienced participants' initial judgements (Round 1), at either the individual ($t(603.011) = -0.274$, $P = 0.784$, $\hat{\beta} = -0.005$, 95%CI = [−0.043, 0.602], TOST: $t(605) = -71.2$, $P < 0.001$; Fig. 4a), or group level ($t(114) = -1.145$, $P = 0.255$, $\hat{\beta} = -0.033$, 95%CI = [−0.09, 0.024], TOST: $t(115) = 6.17$, $P < 0.001$; Fig. 4c). Likewise, error-based

## Table 1 | Replication results

| Original paper | Replication type | RR project | Original sample size | RR sample size | Original effect size | RR effect size | Original inference criteria | RRP value | RR power α=0.05[a] | RR power α=0.025[a] |
|---|---|---|---|---|---|---|---|---|---|---|
| Abouk & Heydari, 2020 | SDR | osf.io/nmzbh | 51 states | 51 states | [b] | [b] | <0.01 | 0.132 | >0.999 | >0.999 |
| Al-Tammemi et al., 2020 | NDR | osf.io/mqvyf | 381 participants | 733 participants | Odds ratio=0.45 | Odds ratio=0.4672 | <0.0001 | <0.001 | 0.995 | 0.989 |
| Bertin et al., 2020 | NDR | osf.io/y5kbe | 409 participants | 59 participants | Cohen's $f^2$=0.2549 | Cohen's $f^2$=0.3219 | <0.001 | <0.001 | 0.900 | 0.837 |
| Bischetti et al., 2020 | NDR | osf.io/n2dhg | 63,036 ratings | 214,696 ratings | [b] | [b] | <0.05 | 0.088 | >0.999 | >0.999 |
| Blagov, 2020 | NDR | osf.io/uzmtn | 502 participants | 450 participants | Partial $r$=0.21 | Partial $r$=0.562 | 0.0156 | <0.001 | 0.918 | 0.866 |
| Carrillo-Vega et al., 2020 | SDR | osf.io/hpgvj | 9,845 patients | 271,427 patients | Odds ratio=1.53 | Odds ratio=1.4273 | <0.001 | <0.001 | >0.999 | >0.999 |
| Columbus, 2020 | NDR | osf.io/8f5by | 401 participants | 314 participants | Cohen's $f^2$=0.0601 | Cohen's $f^2$=0.2898 | <0.001 | <0.001 | 0.963 | 0.934 |
| de la Vega et al., 2020 | SDR | osf.io/dy52c | 64 participants | 700 participants | Pearson's $r$=0.809 | Pearson's $r$=−0.0141[c] | 0.009 | 0.71 | >0.999 | >0.999 |
| Du et al., 2020 | SDR | osf.io/da26y | 64 days | 63 days | Cohen's $f^2$=2.5444 | Cohen's $f^2$=4.5939 | <0.001 | <0.001 | >0.999 | >0.999 |
| Erceg et al., 2020 | NDR | osf.io/z6mkt | 880 participants | 531 participants | [b] | [b] | 95% CI [0.09, 0.18] | <0.001 | 0.982 | 0.966 |
| Flesia et al., 2020 | SDR | osf.io/2px95 | 2,053 participants | 1,844 participants | Cohen's $f^2$=0.0059 | Cohen's $f^2$=0.000042655[c] | 0.0005 | 0.783 | 0.815 | 0.730 |
| Gerhold, 2020 | SDR | osf.io/4rxgz | 1,242 participants | 10,767 participants | Cohen's $d$=0.1464 | Cohen's $d$=0.1668 | <0.01 | <0.001 | >0.999 | >0.999 |
| Goh, 2020 | NDR | osf.io/3czus | 241 participants | 159 participants | Cohen's $f^2$=0.0903 | Cohen's $f^2$=0.0478 | <0.001 | 0.007 | 0.902 | 0.842 |
| González-Marrón & Martínez-Sánchez, 2020[d] | SDR | osf.io/pr5jm | 27 countries | 27 countries | Spearman's $\rho$=−0.476 | Spearman's $\rho$=−0.5047 | 0.012 | 0.007 | 0.451 | 0.335 |
| Hossain, 2020 | SDR | osf.io/h6wg9 | 139 countries | 150 countries | Cohen's $f^2$=0.1148 | Cohen's $f^2$=0.0003 | 0.0001 | 0.826 | 0.944 | 0.903 |
| Imhoff & Lamberty, 2020 | NDR | osf.io/9beuv | 220 participants | 102 participants | Cohen's $f^2$=0.1631 | Cohen's $f^2$=0.1155 | <0.001 | 0.001 | 0.934 | 0.887 |
| Kachanoff et al., 2020 | NDR | osf.io/ytuk9 | 259 participants | 246 participants | [b] | [b] | <0.001 | 0.120 | 0.852 | 0.778 |
| Kavanagh et al., 2020 | SDR | osf.io/gdv4x | 3,037 counties | 3,076 counties | Cohen's $f^2$=0.0048 | Cohen's $f^2$=0.0061 | <0.001 | <0.001 | 0.811 | 0.726 |
| Kuratani, 2020[d] | NDR | osf.io/kbm46 | 78 countries | 103 countries | 25.4% | −79.2446%[c] | 0.029 | 0.14 | 0.065 | 0.034 |
| Malik et al., 2020 | SDR | osf.io/zd2uq | 41 cities | 39 cities | 23% | 26% | 95% CI [20%, 27%] | <0.001 | >0.999 | >0.999 |
| Messner & Payson, 2020[d] | SDR | osf.io/4589g | 1,140 counties | 1,142 counties | [b] | [b,c] | <0.001 | 0.089 | 0.696 | 0.591 |
| Muto et al., 2020 | SDR | osf.io/8uex5 | 8,548 participants | 3,119 participants | Odds ratio=1.635 | Odds ratio=2.1805 | <0.01 | <0.001 | 0.840 | 0.763 |
| Pennycook, McPhetres, Bago et al., 2020 | NDR | osf.io/6ewha | 689 participants | 87 participants | Pearson's $r$=−0.46 | Pearson's $r$=−0.402 | <0.001 | <0.001 | 0.918 | 0.864 |
| Pennycook, McPhetres, Zhang et al., 2020 | NDR | osf.io/rkfq5 | 14,932 ratings | 36,930 ratings | Cohen's $f^2$=0.0009 | Cohen's $f^2$=0.00009 | 0.0003 | 0.07 | 0.999 | 0.997 |
| Pfattheicher et al., 2020 | NDR | osf.io/nv6a3 | 322 participants | 63 participants | Pearson's $r$=0.61 | Pearson's $r$=0.514 | <0.001 | <0.001 | 0.977 | 0.955 |
| Rothgerber et al., 2020 | NDR | osf.io/ky2wm | 573 participants | 322 participants | Pearson's $r$=−0.24 | Pearson's $r$=−0.2090 | <0.001 | <0.001 | 0.904 | 0.846 |
| Sala & Miyakawa, 2020[d] | SDR | osf.io/6zjdh | 136 countries | 162 countries | Cohen's $f^2$=0.0726 | Cohen's $f^2$=0.0017 | 0.0025 | 0.603 | 0.716 | 0.612 |
| Seale et al., 2020 | SDR | osf.io/xr8gz | 1,420 participants | 660 participants | Adjusted odds ratio=5.5 | Adjusted odds ratio=1.0818 | <0.05 | 0.773 | 0.997 | 0.993 |
| Simione & Gnagnarella, 2020[d] | SDR | osf.io/md5pu | 353 participants | 552 participants | Odds ratio=2.72 | Odds ratio=1.6862 | <0.01 | 0.214 | 0.431 | 0.325 |
| Šrol et al., 2020[d] | SDR | osf.io/cqxyh | 783 participants | 288 participants | Pearson's $r$=0.17 | Pearson's $r$=−0.1096[c] | <0.001 | 0.063 | 0.582 | 0.469 |

## Table 1 (continued) | Replication results

| Original paper | Replication type | RR project | Original sample size | RR sample size | Original effect size | RR effect size | Original inference criteria | RRP value | RR power α=0.05[a] | RR power α=0.025[a] |
|---|---|---|---|---|---|---|---|---|---|---|
| Stanley et al., 2020 | NDR | osf.io/cbknq | 278 participants | 2,140 participants | Pearson's r=0.14 | Pearson's r=−0.0210[c] | 0.02 | 0.332 | 0.998 | 0.996 |
| Sternisko et al., 2020 | SDR | osf.io/uscaj | 293 participants | 1,552 participants | Cohen's $f^2$=0.2915 | Cohen's $f^2$=0.3688 | <0.001 | <0.001 | >0.999 | >0.999 |
| Teovanović et al., 2020 | NDR | osf.io/5vgjw | 407 participants | 257 participants | Cohen's $f^2$=0.0578 | Cohen's $f^2$=0.4710 | <0.001 | <0.001 | 0.817 | 0.732 |
| Wise et al., 2020 | NDR | osf.io/fypm4 | 1,152 participants | 1,339 participants | Cohen's $f^2$=0.0158 | Cohen's $f^2$=0.000020622[c] | <0.001 | 0.869 | 0.978 | 0.958 |
| Wissmath et al., 2020 | SDR | osf.io/fjbvp | 1,565 participants | 958 participants | Cohen's $f^2$=0.0625 | Cohen's $f^2$=0.0117 | <0.001 | 0.003 | >0.999 | 0.999 |

Replication results, compared alongside original results ($n$=35); NDR, new data replication; SDR, secondary data replication. Bibliographic details of the original papers can be found in Supplementary Information 5. [a]Indicates power to detect 75% of the original effect size. [b]The original paper did not report a conventional effect size for the selected result, and a conventional effect size could not be calculated from what was reported in the original paper and/or due to the complex nature of the original analysis/design. However, the coefficient and standard error of the selected result were reported in the original paper. Therefore, in these cases, the a priori power calculation for the replication study was based on the original coefficient and standard error, and the original coefficient was compared to the replication study's coefficient to determine whether the replication effect was in the same direction as the original effect. [c]Indicates an effect in the opposite direction of the original effect. [d]Indicates the replication was not deemed to have sufficient statistical power and was not included in further analyses (using α=0.05).

accuracy after discussion (Round 2) did not differ between the two samples, at either the individual ($t$(591) = 0.431, $P$= 0.667, $\hat{\beta}$ = 0.0079, 95%CI = [−0.028, 0.044], TOST: $t$(593) = −56.3, $P$< 0.001), or group level ($t$(113) = −0.58, $P$= 0.563, $\hat{\beta}$ = −0.020, 95%CI = [−0.087, 0.047], TOST: $t$(115) = 9.534, $P$< 0.001). We also did not find evidence of a difference between beginners and more-experienced participants in terms of classification accuracy in Round 1 ($t$(336) = 0.886, $P$= 0.376, $\hat{\beta}$ = 0.033, 95%CI = [−0.041, 0.107], TOST: $t$(337) = −8.06, $P$< 0.001) and only a small difference in Round 2 ($t$(326.925) = 2.131, $P$= 0.034, $\hat{\beta}$ = 0.081, 95%CI = [0.007, 0.156]; Fig. 4b). At the group level, experienced participants correctly classified 69% and beginners correctly classified 59% of the claims they assessed in Round 1. Both performed better on the new data replications (72% for experienced participants, 63% for beginners) than the secondary-data replications (65% for experienced participants, 54% for beginners). In Round 2, classification accuracy was near-identical among experienced (64%) and beginner participants (66%), representing a decrease for the former but an increase for the latter. The experienced participants' classification accuracy was unchanged in Round 2 for new data replications (72%) but lower for secondary-data replications (54%). In contrast, the beginners' classification accuracy improved for secondary-data replications (69%) in Round 2 and remained the same for new data replications (63%).

Despite the similarities between the two groups in overall accuracy, beginners were more likely to shift their best estimates in the right direction after discussion, thus improving their classification accuracy more than experienced participants ($t$(327) = 2.227, $P$= 0.027, $\hat{\beta}$ = 0.056, 95%CI = [0.007, 0.105]). On error-based accuracy, the difference between beginners and more-experienced participants in the magnitude of the shift at both the individual and group level was not significant ($t$(591.044) = 1.646, $P$= 0.100, $\hat{\beta}$ = 0.018, 95%CI = [−0.003, 0.04], TOST: $t$(593) = −69.7, $P$< 0.001 and $t$(113) = 1.405, $P$= 0.163, $\hat{\beta}$ = 0.014, 95%CI = [−0.005, 0.032], TOST: $t$(115) = −40.8, $P$< 0.001, respectively).

**Predictors of accuracy.** We did not find any relationship between classification accuracy and metaresearch experience ($r$(177) = −0.055, $P$= 0.462 and $r$(172) = −0.059, $P$= 0.444 for Rounds 1 and 2, respectively), experience with preregistration ($r$(177) = −0.021, $P$= 0.781 and $r$(172) = −0.022, $P$= 0.770), quiz total scores ($r$(177) = 0.078, $P$= 0.301 and $r$(172) = −0.013, $P$= 0.866), quiz statistic questions ($r$(177) = 0.111, $P$= 0.140 and $r$(172) = 0.004, $P$= 0.959) and quiz replication questions ($r$(177) = −0.007, $P$= 0.925 and $r$(172) = 0.004, $P$= 0.955). While we also did not find a relationship between Round 1 classification accuracy and participants' answers to quiz questions on questionable research practices ($r$(177) = −0.100, $P$= 0.180), we did observe a weak, but significant

correlation in Round 2 ($r$(172) = −0.155, $P$= 0.041). Overall, the participants' characteristics we tested were not reliably correlated with judgement accuracy in this study.

**Accuracy of market prices and comparison between elicitation methods.** Independent of the structured elicitation of predictions about replicability using beginners and experienced samples, another team (T.P., F.H., M.J., Y.L.) assessed the replicability of the claims included in the COVID-19 Preprint Replication Project using a prediction market and simple incentivized surveys. We report on four of the preregistered methods of this team for predicting replicability: the $P$ value-based initial prediction market price, the final prediction market price, the survey means and the means of the survey responses of the five top-ranked forecasters, as determined by the surrogate scoring rule (SSR; see Methods). Average forecast, average error-based accuracy (including 95% CI), classification accuracy and the Spearman correlation between forecast and outcome (including $P$ value) are shown in Table 4.

We investigated the relationship between the predictions of replicability post interaction for both structured groups and the markets for all 100 claims included in this study. We found that the experienced participants were strongly correlated ($r$(98) = 0.68, $P$< 0.001) to the final market prices, whereas beginners were only weakly correlated ($r$(98) = 0.35, $P$= 0.0004; Fig. 5). This difference in correlation between market scores and beginner versus experienced participants was significant ($t$(196) = −2.107, $P$= 0.036, $\hat{\beta}$ = −0.354, 95%CI = [−0.684, −0.025]; see Methods for the full model).

### Behaviour on all claims

These analyses considered individual participant behaviour on all 100 claims (see Supplementary Information 4 for beginner and experienced participants' predictions on all 100 claims). Over the entire dataset of 100 claims, experienced participants' and beginners' group-level Round 2 predictions were moderately correlated ($r$(98) = 0.48, $P$< 0.001).

**Best estimates of replication probability.** Beginners had higher best estimates than experienced participants in both Round 1 ($t$(1,981.3591) = 4.596, $P$< 0.0001, $\hat{\beta}$ = 0.038, 95%CI = [0.022, 0.054]) and Round 2 ($t$(1,980.0922) = 8.008, $P$< 0.0001, $\hat{\beta}$ = 0.055, 95%CI = [0.041, 0.068]; see Fig. 6).

Further examination of the absolute magnitude of the shift in individual probability estimates between rounds showed that beginners made a significantly larger shift (by ~5 points) compared with experienced participants ($t$(1,989.6550) = 10.80, $P$< 0.0001, $\hat{\beta}$ = 0.0494, 95%CI = [0.04, 0.058]), confirming our preregistered hypothesis.

**Table 2 | Summary of demographic and expertise characteristics of participants in the two conditions**

| | High task expertise (experienced participants) | Low task expertise (beginners) | Inferential test of difference between experienced participants and beginners |
|---|---|---|---|
| Age (mean(s.d.)) | 31.9 (9) ($n$=99) | 21.9 (3.2) ($n$=86) | $t$(183)=9.733, $P$<0.001, 95%CI=[7.944,11.984] |
| Gender | 52 Female, 44 Male ($n$=96) | 49 Female, 36 Male ($n$=85) | $\chi^2$(1)=0.221, $P$=0.638 |
| Education (highest attained) | 25 Doctorate, 43 Masters, 28 Undergraduate, 1 Professional degree, 0 Associate degree, 1 High school ($n$=98) | 0 Doctorate, 0 Masters, 18 Undergraduate, 0 Professional degree, 6 Associate degree, 58 High school ($n$=82) | |
| Region (based on first-listed nationality) | 21 Asia 0 Africa 28 Europe 27 North America 21 Oceania 0 Central and South America ($n$=97) | 5 Asia 1 Africa 4 Europe 68 North America 1 Oceania 5 Central and South America ($n$=84) | |
| Degree discipline (highest attained) | 58 Social science, 27 Science, 1 Arts/Humanities, 0 Business Admin., 2 Engineering, 11 Other ($n$=99) | 20 Social science, 20 Science, 8 Arts/Humanities, 4 Business Admin., 0 Engineering, 10 Other ($n$=62) | |
| Number of courses taken (statistics and/or quantitative research methods, max. 6) (mean(s.d.)) | 4.4 (1.7) ($n$=97) | 1.7 (1.2) ($n$=86) | $t$(181)=12.607, $P$<0.0001, 95%CI=[2.320,3.182] |
| Number of publications (mean(s.d.)) | 8 (11.6) ($n$=96) | 0.1 (0.4) ($n$=84) | $t$(178)=6.201, $P$<0.0001, 95%CI=[5.347,10.339] |
| Career stage | 2 Undergraduate, 57 Graduate student, 16 Early-career, 16 Mid-career, 5 Senior, 2 Other ($n$=98) | 87 Undergraduate ($n$=87) | |
| Technical expertise: math (self-reported) | 2.653 (0.740); md=2 ($n$=98) | 2.353 (0.972); md=2 ($n$=85) | $t$(181)=0.692, $P$=0.490, 95%CI difference [−0.338, 0.162] |
| Technical expertise: modelling (self-reported) | 1.742 (1.201); md=2 ($n$=97) | 1.302 (1.096); md=1 ($n$=86) | $t$(181)=2.571, $P$=0.011, 95%CI difference [0.102,0.776] |
| Technical expertise: statistics (self-reported) | 2.808 (0.738); md=3 ($n$=99) | 1.686 (0.997); md=1 ($n$=86) | $t$(183)=8.771, $P$<0.001, 95%CI difference [0.870,1.375] |
| Technical expertise: probability (self-reported) | 2.1934 (0.821); md=2 ($n$=98) | 1.588 (0.955); md=1 ($n$=85) | $t$(181)=4.616, $P$<0.001, 95%CI difference [0.347,0.865] |
| Technical expertise: experimental design (self-reported) | 2.7273 (0.924); md=3 ($n$=99) | 1.814 (1.213); md=2 ($n$=86) | $t$(183)=5.802, $P$<0.001, 95%CI difference [0.603,1.224] |
| Technical expertise: risk analysis (self-reported) | 0.929 (1.023); md=1 ($n$=99) | 1.151 (1.068); md=1 ($n$=86) | $t$(183)=0.573, $P$=0.568, 95%CI difference [−0.418,0.230] |
| Technical expertise: forecasting (self-reported) | 0.727 (1.048); md=0 ($n$=99) | 0.907 (1.025); md=1 ($n$=86) | $t$(183)=2.260, $P$=0.025, 95%CI difference [−0.601,−0.041] |
| Previous experience with replication studies (checkbox) | 34 indicated yes ($n$=99) | 6 indicated yes ($n$=88) | $\chi^2$(1)=20.99, $P$<0.0001 |
| Previous experience with preregistration (checkbox) | 38 indicated yes ($n$=99) | 5 indicated yes ($n$=88) | $\chi^2$(1)=28.14, $P$<0.0001 |
| Previous experience with metaresearch (checkbox) | 27 indicated yes ($n$=99) | 1 indicated yes ($n$=88) | $\chi^2$(1)=25.00, $P$<0.0001 |
| Quiz score (max. 22 points) (mean(s.d.)) | 11.9 (4) ($n$=98) | 6.6 (2.7) ($n$=87) | $t$(178)=10.589, $P$<0.0001, 95%CI=[4.325,6.305] |

Age and number of publications were elicited as a range, with midpoints used for descriptive statistics. Means and standard deviations of range midpoints are reported here. The $N$ varies due to missing data on some of the demographic variables.

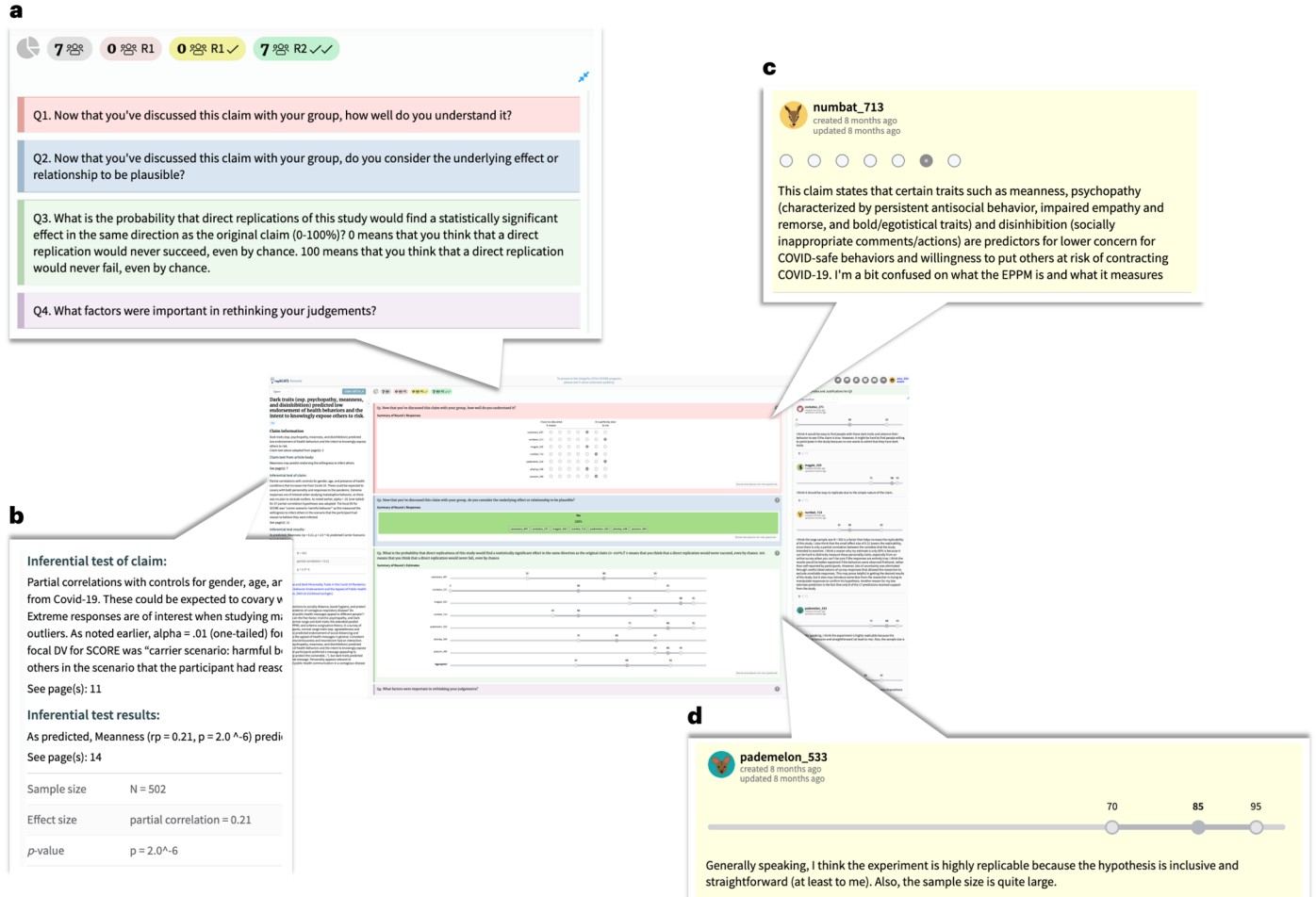

**Fig. 2 | Overview of the repliCATS platform.** Overview of the repliCATS platform as displayed to participants in Round 2. The full platform view is shown in the centre, summarizing Round 1 responses from 7 participants for one of the evaluated research claims. Enlarged platform components show examples of the Round 2 elicitation questions (collapsed, **a**), the research claim's statistical summary information (**b**), an example restatement of the claim from one of the participants in response to Q1 on the platform (**c**) and an example of Round 1 participant reasoning paired with their quantitative replicability judgement in response to Q3 on the platform (**d**).

**Judgement uncertainty.** The uncertainty in participants' judgements is expressed in the interval width (that is, the distance between the lower and upper bound), with wider intervals indicating greater uncertainty around the replication probability estimate. Interval widths were similar between the two groups in Round 1 ($t(1988.3245) = -1.044$, $P = 0.297$, $\hat{\beta} = -0.008$, 95%CI = [−0.023, 0.007], TOST: $t(2,079) = 252.689$, $P < 0.001$), but beginners had significantly narrower intervals in Round 2 ($t(1,985.2034) = 4.882$, $P < 0.0001$, $\hat{\beta} = 0.029$, 95%CI = [0.041, 0.017]). Beginners and experienced participants differed significantly with respect to how they adjusted their interval widths after discussion ($t(1,987.6112) = 6.617$, $P < 0.0001$, $\hat{\beta} = 0.0318$, 95%CI = [0.022, 0.041]), further supporting our preregistered hypothesis.

Probability estimates of binary variables (will replicate vs will not replicate) are most uncertain when they are very close to 0.5. Therefore, we also considered all best estimates larger than or equal to 0.4 and less than or equal to 0.6 (called this the middle bin) and investigated how often participants updated their judgements outside of this interval after discussion. Only ~2% of all Round 1 best estimates fell outside the middle bin (2% for experienced and 2.3% for beginner participants). However, after discussion, notably more beginners updated at least one of their best estimates to a value outside the middle bin (70% of beginners and 46% of experienced participants). Moreover, while only 13% of experienced participants moved out of the middle bin more than half of their Round 1 best estimates, 36% of beginners did so.

## Discussion

We elicited judgements about the replicability of social science research on the COVID-19 pandemic drawn from 100 preprints and conducted 29 high-powered new replications. Participants' average estimates using a Delphi-style structured protocol were in line with previously observed replication rates in the social sciences, that is, ~65.4% and were largely consistent with previous studies on forecasting replicability of published results. We also directly compared participants' performance with lower task expertise to a more-experienced sample. We construed the difference between the two samples in terms of their experience with judging the quality of empirical research in the social and behavioural sciences, and indeed, they differed significantly on multiple dimensions, including number of publications, number of courses in statistics/quantitative methodology taken, statistical knowledge (both self-assessed and assessed through a quiz) and in terms of how their judgements correlated with those of participants to a prediction market with previous forecasting experience. Moreover, the experienced participants were drawn from the same population that academic journal editors often recruit their peer-reviewers from to judge the quality of submitted research. Despite these differences, we ultimately did not find evidence of a difference in their predictive ability, and final judgements (at the end of the Delphi-style process) were correlated, with both groups achieving above-chance accuracy on most metrics. Using the TOST method, we estimate that the true difference in raw means between beginners and experienced participants is within ±0.05 points.

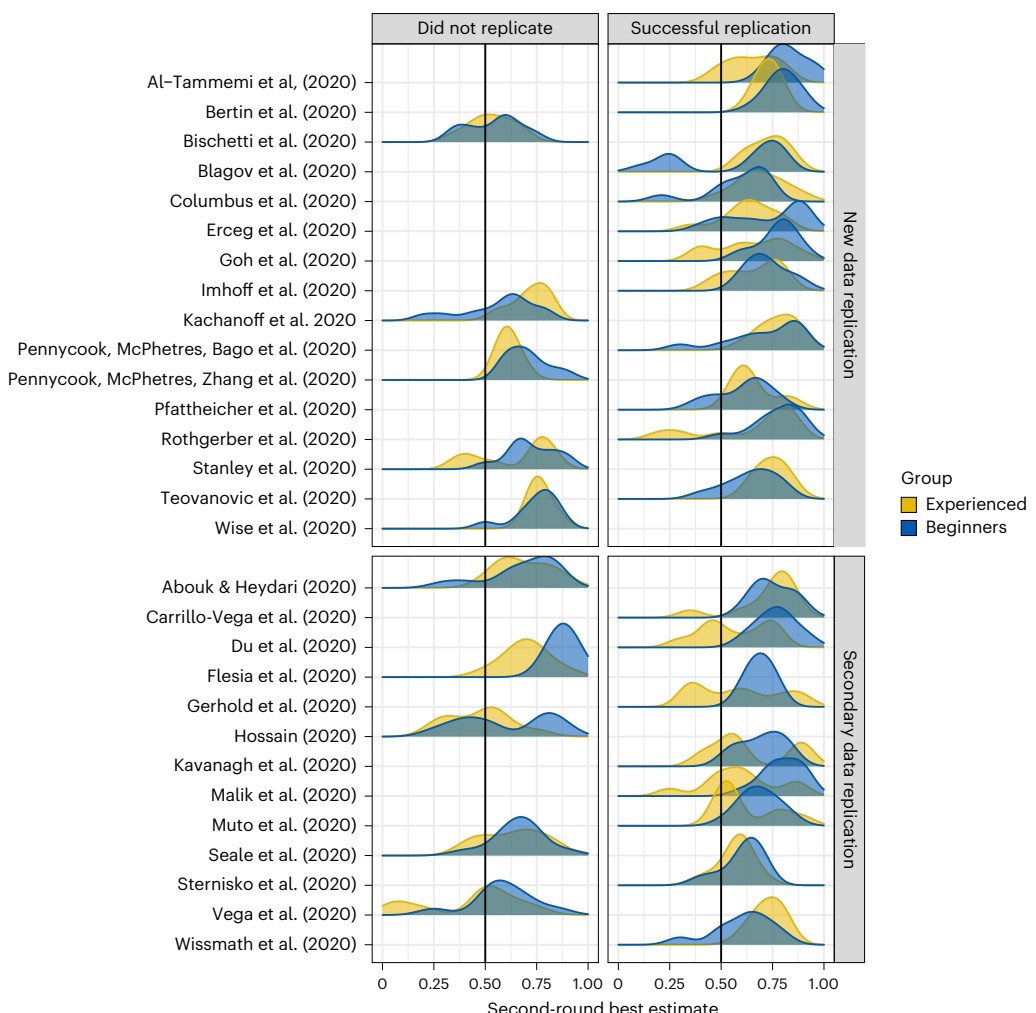

**Fig. 3 | Participants' best estimates.** Smoothed distribution of participants' best estimates for each of the 29 known-outcome research claims with ≥0.8 power with an $\alpha$ = 0.05, organized by type of replication (new or secondary data) and success (did or did not replicate). Experienced participants are shown in yellow and beginners in blue.

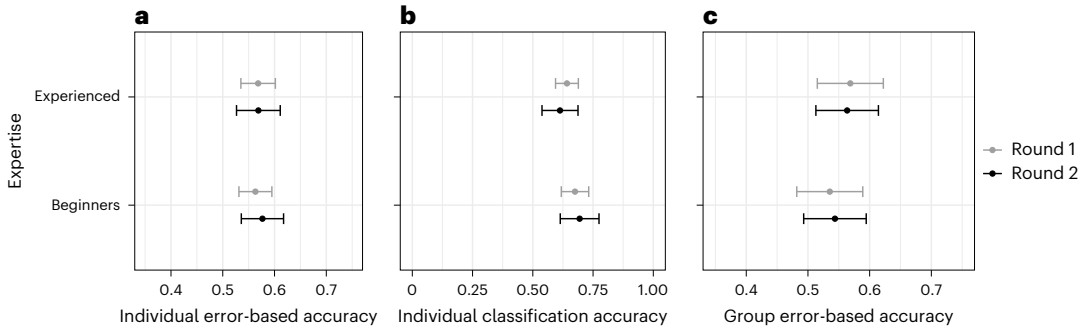

**Fig. 4 | Predictive accuracy results.** Average error-based and classification accuracy results and 95% confidence intervals for both individuals and groups of beginners and experienced participants. Estimates and 95% confidence intervals (mean ± s.e. ×1.96) drawn from linear models described in Table 3 refit with no reference class. Statistical test results: **a**, Round one (Beginners: estimated effect size $\hat{\beta}$ = 0.563, 95%CI = [0.531, 0.595]; Experienced: $\hat{\beta}$ = 0.568, 95%CI = [0.535, 0.602]; Difference: $t$(603.01) = −0.274, $P$ = 0.784, $\hat{\beta}$ = −0.005, 95%CI = [−0.043, 0.032], $n$ = 606); Round two (Beginners: $\hat{\beta}$ = 0.577, 95%CI = [0.536, 0.617]; Experienced: $\hat{\beta}$ = 0.569, 95%CI = [0.527, 0.611]; Difference: $t$(591.00) = 0.431, $P$ = 0.667, $\hat{\beta}$ = 0.008, 95%CI = [−0.028, 0.044], $n$ = 594). **b**, Round one (Beginners:

$\hat{\beta}$ = 0.675, 95%CI = [0.618, 0.732]; Experienced: $\hat{\beta}$ = 0.642, 95%CI = [0.594, 0.689]; Difference: $t$(336.00) = 0.886, $P$ = 0.376, $\hat{\beta}$ = 0.033, 95%CI = [−0.041, 0.107], $n$ = 338); Round two (Beginners: $\hat{\beta}$ = 0.694, 95%CI = [0.614, 0.775]; Experienced: $\hat{\beta}$ = 0.613, 95%CI = [0.538, 0.688]; Difference: $t$(326.93) = 2.131, $P$ = 0.034, $\hat{\beta}$ = 0.081, 95%CI = [0.007, 0.156], $n$ = 329). **c**, Round one (Beginners: $\hat{\beta}$ = 0.535 = 0.535, 95%CI = [0.482, 0.589]; Experienced: $\hat{\beta}$ = 0.569, 95%CI = [0.515, 0.622]; Difference: $t$(114.00) = −1.145, $P$ = 0.255, $\hat{\beta}$ = −0.033, 95%CI = [−0.09, 0.024], $n$ = 116); Round two (Beginners: $\hat{\beta}$ = 0.544, 95%CI = [0.493, 0.594]; Experienced: $\hat{\beta}$ = 0.564, 95%CI = [0.513, 0.614]; Difference: $t$(113.00) = −0.580, $P$ = 0.563, $\hat{\beta}$ = −0.020, 95%CI = [−0.087, 0.047], $n$ = 116).

**Table 3 | List of models used**

| Description | Data subset | Response variable | Fixed effects | Random intercept | Preregistered analysis? |
|---|---|---|---|---|---|
| Effect of expertise on second-round individual-level accuracy | Replicated papers only | Round-two individual error-based accuracy | Expertise | Replication type | No |
| Effect of expertise on second-round individual-level accuracy | Replicated papers only | Round-two individual classification accuracy | Expertise | Replication type | No |
| Effect of expertise on second-round group accuracy | Replicated papers only | Round-two group error-based accuracy | Expertise | Replication type[a] | No |
| Effect of expertise on first-round individual-level accuracy | Replicated papers only | Round-one individual error-based accuracy | Expertise | Replication type | Yes |
| Effect of expertise on first-round individual-level accuracy | Replicated papers only | Round-one individual classification accuracy | Expertise | Replication type | No |
| Effect of expertise on first-round group accuracy | Replicated papers only | Round-one group error-based accuracy | Expertise | Replication type[a] | No |
| Effect of expertise on the shift in accuracy between rounds | Replicated papers only | Shift in individual error-based accuracy between rounds | Expertise | Replication type[a] | No |
| Effect of expertise on the shift in classification accuracy between rounds | Replicated papers only | Shift in individual classification accuracy between rounds | Expertise | Replication type | No |
| Effect of expertise on the shift in group accuracy between rounds | Replicated papers only | Shift in group error-based accuracy between rounds | Expertise | Replication type[a] | No |
| Effect of expertise on round-one interval width | All papers | Round-one interval width | Expertise | Paper ID | No |
| Effect of expertise on round-two interval width | All papers | Round-two interval width | Expertise | Paper ID | No |
| Effect of expertise on the shift in interval width | All papers | Shift in interval width between rounds | Expertise | Paper ID | No |
| Effect of expertise on the shift in best estimates | All papers | Shift in best estimates between rounds | Expertise | Paper ID | Yes |
| Is there a difference in error-based accuracy between the two beginner groups? | Replicated papers only | Round-two individual error-based accuracy | Group | Replication type | No |
| Is there a difference in classification accuracy between the two beginner groups? | Replicated papers only | Round-two individual classification accuracy | Group | - | No |
| Effect of expertise on shift in estimated replication rate (raw shift values) | All papers | Shift in best estimates between rounds | Expertise | Paper ID | No |
| Is there an effect of interval width on accuracy? | Replicated papers only | Round-one individual error-based accuracy | Round one interval width | Replication type | No |
| Is there an effect of interval width on accuracy? | Replicated papers only | Round-two individual error-based accuracy | Round two interval width | Replication type | No |
| Effect of expertise on round-one best estimates | All papers | Round-one best estimates | Expertise | Paper ID | No |
| Effect of expertise on round-two best estimates | All papers | Round-two best estimates | Expertise | Paper ID | No |
| Effect of expertise on relationship with Markets' estimates | Replicated papers only | Markets' scores | repliCATS confidence scores + Expertise + Interaction term | - | No |

[a]Random intercept terms excluded from $α = 0.025$ analyses due to reduced dataset size.

The above results confirm our preregistered hypothesis regarding the behaviour of beginners and experienced participants but fail to confirm our hypothesis regarding their accuracy. We hypothesized that experienced participants would outperform beginners. The rationale for this hypothesis is based on the fact that the experienced sample, composed primarily of doctoral students and early-career researchers, would have had many hours of deliberate practice[37] with judging the quality of research and the plausibility of findings. These are presumed to be learned skills honed by sustained feedback from professors, supervisors, mentors, colleagues, referees and conference audiences. Beginner samples of undergraduate students should have had much less opportunity to develop these skills. Several choices in the design

of this study make it difficult to interpret this result. First, our experienced participants did not have a strong claim to domain expertise in the forecasting questions they answered. Indeed, it is unlikely that there are any experts who can competently answer questions about 'COVID-19 humour in the Italian population during the lockdown', 'US political affiliation impact on social distancing' and 'COVID-19 severity and lethality in older Mexican adults'. While experienced participants were markedly better on various dimensions of methodological expertise, some of which have been previously correlated with forecasting accuracy[38], this might not have been enough to lift their forecasting performance above that of beginners, given the diversity of research topics they were asked to assess.

**Table 4 | Results from the replication markets project**

| Assessment method | Average forecast | Average error-based accuracy [95% CI] | Average classification accuracy | Correlation between forecast and outcome | P value |
|---|---|---|---|---|---|
| *P* value-based | 0.60 | 0.64 [0.56–0.73] | 0.76 | 0.47 | 0.01 |
| Final market price | 0.62 | 0.61 [0.53–0.68] | 0.69 | 0.41 | 0.03 |
| Survey means | 0.65 | 0.61 [0.53–0.69] | 0.72 | 0.49 | 0.008 |
| Surveys, SSR aggregated | 0.66 | 0.65 [0.55–0.74] | 0.76 | 0.41 | 0.03 |

Second, while we attempted to replicate all studies in our corpus using new data (and indeed, our elicitation questions explicitly asked participants for their judgements on the probability of a successful direct replication of the claim under consideration), only a subset of our replications lived up to that expectation. Both groups performed better and were more accurate when predicting replications based on new data than secondary data replications, with experienced participants outperforming beginners; the former correctly classified 72% of new data replications, while the latter correctly classified only 63% of claims.

The similarity in prediction accuracy between the two groups may be due to the high level of uncertainty inherent in assessing the quality of preprints about an evolving public health crisis. Preprints differ in numerous ways from journal articles that might influence judgements of replicability. They have not yet passed peer-review scrutiny, which would be expected, at the very least, to improve the clarity of reporting and potentially enhance the quality of the manuscript. In their comparison of COVID-19 preprints and journal publications, ref. 39 concluded that the publication pipeline had minimal but beneficial effects on manuscripts by, for example, increasing sample sizes, improving statistical analysis and reporting, and making author language more conservative (that is, less likely to overstate importance or significance of results or oversell novelty). This may have differentially affected experienced participants who were more likely to have developed heuristics that use that information.

At the same time, the idea of 'replicability' itself may be harder for participants to parse in a fast-evolving crisis like the COVID-19 pandemic (especially at the time when data were collected for this study), leading to conceptual slippage: participants may have folded in the concept of generalizability (or conflated direct and conceptual replication), which is much harder to predict. The replication

studies themselves reflect a different point in time, as the social and behavioural impacts of the pandemic evolved throughout the past 4 years, making it challenging to retest how, for example, political ideology in 2020 influenced one's decision to social distance. This makes 'social and behavioural fast science' very different from 'medical fast science'.

Our hypothesis that beginners would make more substantive changes to their judgements after receiving feedback on their initial assessment was supported. On average, they shifted their probability judgements by ~5 points more than experienced participants after conferring with their peers, and they became more confident in their judgements. For the 29 high-powered replication outcomes, beginners tended to shift their judgement in the right direction after discussion, whereas experienced participants shifted very little. This suggests that beginners may benefit more from discussion because individual group members may not hold sufficient knowledge to make a well-reasoned judgement or may be unduly influenced by one aspect of the study. Sharing information through discussion may allow them to make a more informed decision. Experienced participants, on the other hand, may have believed (rightly or not) in the extent of their knowledge or may have been more entrenched in making judgements about the quality of research using tried and tested heuristics. Beginners were also more likely than experienced participants to update their judgements away from the [0.40–0.60] range, suggesting that they were open to more 'extreme' shifts, possibly because they were more willing to truly take on board novel information even if it diverged substantially from their initial position, as suggested by their increased confidence post discussion. However, the strength of evidence about this behavioural distinction between the two groups is modest. Our explanation of these behavioural patterns remains speculative, and more investigation is required to verify the extent to which the underlying decision-making processes of beginners and experienced participants predicting the replicability of research claims differ.

This study showed that both beginners and more-experienced participants using a structured process have some ability to make better-than-chance predictions about the reliability of 'fast science' under conditions of high uncertainty. However, performance using both a Delphi method and a prediction market approach was below expectations and below what would be desirable in practice. It is the case, nevertheless, that these contexts are precisely those where we must rely on human judgement (rather than having time to conduct new high-powered replications) and where individuals of varying degrees of both methodological and domain expertise will be called to make critical decisions. Therefore, more research is required to understand who the right experts are in forecasting the replicability

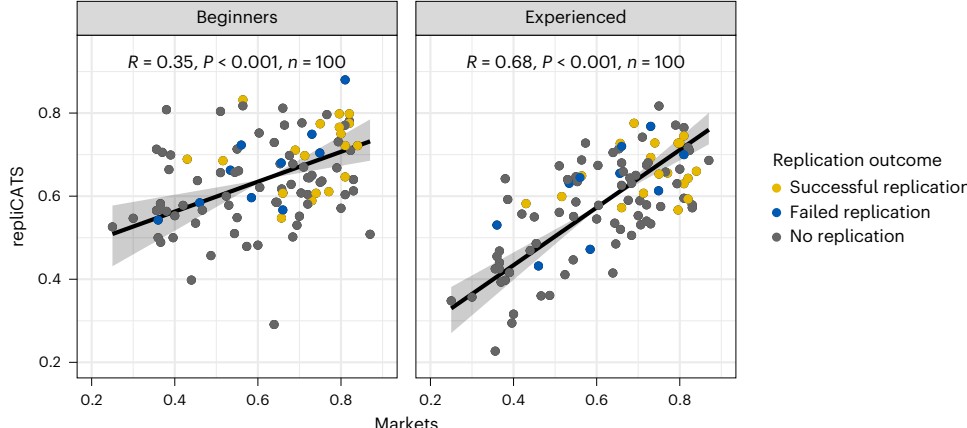

**Fig. 5 | Structured group judgements vs final market prices.** Pearson correlations between Round 2 structured group judgements (collected by the repliCATS team) and final market price for both beginners and experienced

participants. Correlations are calculated with a sample size of 100, and the regression line and 95% confidence intervals are calculated using major axis regression.

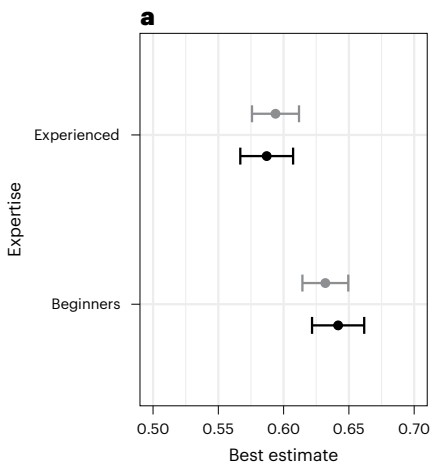

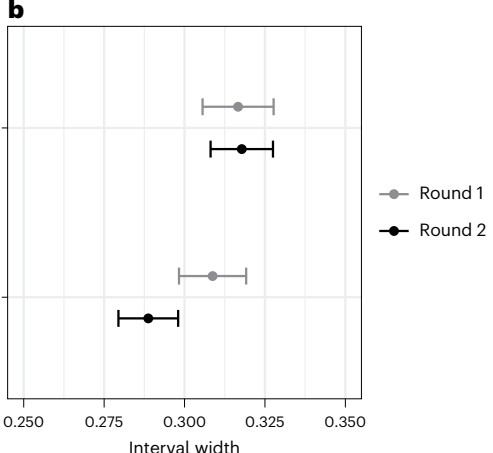

**Fig. 6 | Participants' best estimates and interval widths.** Average best estimates and interval widths for both beginners and experienced participants. Estimates and 95% confidence intervals (mean ± s.e. ×1.96) drawn from linear models described in Table 3 refit with no reference class. Statistical test results: **a**, Round one [Beginners: β̂ = 0.632, 95%CI = [0.614, 0.65]; Experienced: β̂ = 0.594, 95%CI = [0.576, 0.612]; Difference: $t(1981.3591)$ = 4.596, $P$ < 0.0001, β̂ = 0.038, s.e. = 0.008, 95%CI = [0.022, 0.054], $n$ = 2080]; Round two [Beginners: β̂ = 0.642, 95%CI = [0.622, 0.662]; Experienced: β̂ = 0.587, 95%CI = [0.567, 0.607]; Difference:

$t(1980.0922)$ = 8.008, $P$ < 0.0001, β̂ = 0.055, s.e. = 0.007, 95%CI = [0.041, 0.068], $n$ = 2080]. **b**, Round one [Beginners: β̂ = 0.309, 95%CI = [0.298, 0.319]; Experienced: β̂ = 0.317, 95%CI = [0.306, 0.328]; Difference: $t(1988.3245)$ = −1.044, $P$ = 0.297, β̂ = −0.008, s.e. = 0.008, 95%CI = [−0.023, 0.007], $n$ = 2080]; Round two [Beginners: β̂ = 0.289, 95%CI = [0.279, 0.298]; Experienced: β̂ = 0.318, 95%CI = [0.308, 0.327]; Difference: $t(1985.2034)$ = −4.882, $P$ < 0.0001, β̂ = −0.029, s.e. = 0.006, 95%CI = [−0.041, −0.017], $n$ = 2080].

of studies in rapidly growing evidence bases, how their judgements ought to be elicited and how we can better support laypeople to assess scientific evidence about an emerging crisis more accurately.

There are several opportunities for improvement in the methodology of large-scale replication projects such as this in the future. For example, we emulated a decision process from ref. 7 for assessing the outcomes and then continuing data collection for findings that were not statistically significant to improve the efficiency of participant resources and increase power to detect underestimated effects (see Methods for more details). This approach modestly increases the likelihood of false positives. Applying sequential analysis methods would help to mitigate increases in false positive risk[40]. Likewise, given the explicit goal of replicating directional claims, the power to detect

those effects could be increased by a priori adopting one-tailed versus two-tailed designs. We investigated the consequences of only selecting research claims with ≥0.8 power with $\alpha$ = 0.025 for the analyses reported in this paper. We note that when adopting $\alpha$ = 0.025, the number of replications that meet the power threshold drops to 24 (14 new data and 10 secondary-data). In this subcorpus, experienced participants correctly classified 63% of claims in Round 2, and both beginners using our structured elicitation protocol and the final market price achieved 71% classification accuracy. We report the comparisons between different modelling choices in our results in Table 5 (full results are included in Supplementary Information 3). We note that adopting $\alpha$ = 0.025 further diminishes the differences between experienced and beginner participants.

## Table 5 | Model results at two different $\alpha$-levels

| Description | $\alpha$=0.05 result | $\alpha$=0.025 result |
|---|---|---|
| Effect of expertise on second-round individual-level error-based accuracy | $t(591.00)$=0.431, $P$=0.667, $\hat{\beta}$=0.008, 95%CI=[−0.028, 0.044], TOST: $t(593)$=−56.3, $P$<0.001 | $t(486)$=0.556, $P$=0.579, $\hat{\beta}$=0.011, 95%CI=[−0.028, 0.05], TOST: $t(488)$=−43.9, $P$<0.001 |
| Effect of expertise on second-round individual-level classification accuracy[a] | $t(326.93)$=2.131, $P$=0.034, $\hat{\beta}$=0.081, 95%CI=[0.007, 0.156] | $t(298.37)$=1.862, $P$=0.064, $\hat{\beta}$=0.082, 95%CI=[−0.004, 0.169] |
| Effect of expertise on second-round group error-based accuracy | $t(113.00)$=−0.580, $P$=0.563, $\hat{\beta}$=−0.020, 95%CI=[−0.087, 0.047], TOST: $t(115)$=9.534, $P$<0.001 | $t(94)$=−0.736, $P$=0.463, $\hat{\beta}$=−0.026, 95%CI=[−0.097, 0.044], TOST: $t(95)$=6.44, $P$<0.001 |
| Effect of expertise on first-round individual-level error-based accuracy | $t(603.01)$=−0.274, $P$=0.784, $\hat{\beta}$=−0.005, 95%CI=[−0.043, 0.032], TOST: $t(605)$=−71.2, $P$<0.001 | $t(496.00)$=−0.202, $P$=0.840, $\hat{\beta}$=−0.004, 95%CI=[−0.045, 0.036], TOST: $t(498)$=−58.5, $P$<0.001 |
| Effect of expertise on first-round individual-level classification accuracy | $t(336.00)$=0.886, $P$=0.376, $\hat{\beta}$=0.033, 95%CI=[−0.041, 0.107], TOST: $t(337)$=−8.06, $P$<0.001 | $t(305.91)$=1.135, $P$=0.257, $\hat{\beta}$=0.050, 95%CI=[−0.036, 0.136], TOST: $t(307)$=0.022, $P$=0.509 |
| Effect of expertise on first-round group error-based accuracy | $t(114.00)$=−1.145, $P$=0.255, $\hat{\beta}$=−0.033, 95%CI=[−0.09, 0.024], TOST: $t(115)$=6.17, $P$<0.001 | $t(94)$=−1.235, $P$=0.22, $\hat{\beta}$=−0.038, 95%CI=[−0.097, 0.022], TOST: $t(95)$=4.027, $P$<0.001 |
| Effect of expertise on the shift in error-based accuracy between rounds | $t(591.04)$=1.646, $P$=0.100, $\hat{\beta}$=0.018, 95%CI=[−0.003, 0.04], TOST: $t(593)$=−69.7, $P$<0.001 | $t(487)$=1.497, $P$=0.135, $\hat{\beta}$=0.018, 95%CI=[−0.006, 0.042], TOST: $t(488)$=−57.3, $P$<0.001 |
| Effect of expertise on the shift in classification accuracy between rounds[a] | $t(327.00)$=2.227, $P$=0.027, $\hat{\beta}$=0.056, 95%CI=[0.007, 0.105] | $t(298.94)$=1.436, $P$=0.152, $\hat{\beta}$=0.040, 95%CI=[−0.015, 0.096], TOST: $t(300)$=−5.87, $P$<0.001 |
| Effect of expertise on the shift in group error-based accuracy between rounds | $t(113.00)$=1.405, $P$=0.163, $\hat{\beta}$=0.014, 95%CI=[−0.005, 0.032], TOST: $t(115)$=−40.8, $P$<0.001 | $t(94)$=1.008, $P$=0.316, $\hat{\beta}$=0.011, 95%CI=[−0.01, 0.033], TOST: $t(95)$=−34.6, $P$<0.001 |

[a]Indicates a change in significance between the results for different $\alpha$-levels

Moreover, future research can take advantage of the continuous innovation in methods for evaluating replication success, both for cases in which it is relevant to judge single experimental outcomes, as in this case, and for aggregating cumulative evidence to increase precision and estimate the reliability and heterogeneity of effects[41].

Finally, care should be taken when generalizing these findings. Beginners were all recruited from the undergraduate population at a US university. It is valuable to understand how accurately this community, who we might realistically call upon to participate in research assessments[42], can evaluate the credibility of social and behavioural science research. They will also have probably received training similar to that of many policymakers, or journalists who are routinely asked to make such assessments for rapid policy implementation and reporting purposes. Still, we cannot generalize to other groups of similar students from different universities, degree courses and countries, or the general public. We also acknowledge that our preprint selection process was not designed to deliver a representative sample of all COVID-19 social and behavioural science research. Instead, we aimed to provide a suitable testbed for assessing our participants' predictions about the studies' replicability in ways that would allow us to discriminate between poor and good performance. For these reasons, this study should not be interpreted as making any claim about the quality of social and behavioural science on the COVID-19 pandemic.

## Methods

### COVID-19 Preprint Replication Project

**Selecting preprints.** The 100 claims were extracted from preprints that made quantitative claims based on social-behavioural data related to COVID-19. These manuscripts were identified from the preprint servers PsyArXiv (https://psyarxiv.com/), SocArXiv (https://osf.io/preprints/socarxiv) and medRxiv (https://www.medrxiv.org/) through searches on 3, 5 and 13 April, and 8, 18, 26, 27 and 29 May 2020, using the following search terms: "COVID-19", "coronavirus" and "2019-ncov" for PsyArXiv; "COVID-19" for SocArXiv; and the "COVID-19 SARS-CoV-2" (https://connect.medrxiv.org/relate/content/181) and "epidemiology" collection (https://www.medrxiv.org/collection/epidemiology) on medRxiv. Preprints that included an abstract were manually reviewed by a single person in the order that they appeared in the search output on each search date. We continued to search and assess preprints for inclusion until we obtained 100 eligible preprints for this study.

The selection procedure was based first on a quick review of the title and abstract to assess whether the preprint met the following criteria:

- It reported a prediction, explanation or description of human behaviour, where human behaviour is defined at any level of human organization (for example, the individual person, family, political entity, firm, economic unit).
- It used inferential statistics, as opposed to only offering descriptive estimates (for example, an estimate of the peak infection percent in a given population) and reported at least one significant test result.

If the preprint passed these initial screens, it was reviewed to determine whether it contained a claim that could feasibly be replicated in a good faith attempt by November 2020. Replication in this context meant applying the same analytical approach with new data that could be collected or found, such as seeing whether the claim holds in the same region at a later date or a different region or situation in which the claim was expected to occur. This was a subjective assessment as the actual feasibility of conducting a replication was dependent on other factors that were not known at the time of selection, such as the identification of a researcher with the appropriate expertise to conduct the replication or whether data that were assumed to be available in the future would be available.

**Table 6 | Summary of original claims (replicated subset, n = 35)**

| Geographic context | United States: n=15 (43%)<br>Europe: n=11 (31%)<br>Global: n=4 (11%)<br>Asia-Pacific: n=2 (6%)<br>Latin America: n=1 (3%)<br>Middle East: n=1 (3%)<br>Unknown: n=1 (3%) |
|---|---|
| Preprint service | PsyArXiv: n=22 (63%)<br>medRxiv: n=13 (37%) |
| Month posted (2020) | March: n=10 (29%)<br>April: n=12 (34%)<br>May: n=13 (37%) |
| Topic | Psychological correlates of attitudes and behaviours: n=12 (34%)<br>Conspiracy beliefs and misinformation: n=7 (20%)<br>Demographic correlates of attitudes, behaviours and outcomes: n=7 (20%)<br>Social context of behaviours and outcomes: n=5 (14%)<br>Policy analysis: n=4 (11%) |
| Units of analysis[a] | Individuals: n=26 (74%)<br>Geographic units: n=8 (23%)<br>Days: n=1 (3%) |

[a]Of the 26 original claims in which individuals were the primary unit of analysis, 16 were replicated with new data and 10 with secondary data. The remaining claims (n=9) were all replicated with secondary data.

Of the 174 preprints assessed, 113 focused on human behaviour, used inferential tests and were considered replicable in the required timeline. Of these 113 preprints, 100 were included in the final dataset, with the remaining manuscripts discarded for use in this study (for example, used in pilots to refine the process or held in reserve as overage).

**Extracting claims from preprints.** For each preprint reviewed for the sample set, the coder (1) determined the preprint's eligibility; (2) selected one focal result corresponding to an original research claim in the abstract; (3) identified key elements of the claim explaining what was tested, how it was tested and what was found; and (4) tagged and annotated each piece of claim information. A reviewer then assessed the extracted information for comprehensibility and internal consistency, after which the claim information was finalized and sent to other teams for prediction. A summary of original claims is provided in Table 6.

For each eligible article, the coder selected seven pieces of information anchored on a single statistically significant statistical inference:

- Claim 2: An original finding from the abstract supported by the statistical inference.
- Claim 3a: A hypothesis statement or summary of the finding from the main text that aligned with both the Claim 2 statement and the inference.
- Claim 3b: A summary of the methodology that generated the inference.
- Claim 4: The statistical output associated with the inference and (most often) the authors' conclusion or interpretation of that evidence.
- Sample size: If available, the sample size of the analysis that produced the selected inference.
- *P* value: If available, the *P* value of the inference.
- Effect size: If available, the portion of the statistical output closest to a standardized effect size.

**Powering replications.** COVID claims selected by an interested research team for potential replication received a target sample size

determined from an a priori power analysis, which was conducted in R and based primarily on the statistical results reported directly in the original paper. Much of the R code uses the pwr package (v.1.2-2)[43] as well as other packages and approaches to estimating power when necessary. Each power calculation and sample size target was documented in the study's preregistration.

In general, three sample sizes were calculated for each replication attempt, using a two-tailed $\alpha = 0.05$: (1) the sample size required to achieve 90% power to detect 75% of the original effect size (stage 1 sample size); (2) the sample size required to achieve 90% power to detect 50% of the original effect size (stage 2 sample size); and (3) the sample size required to achieve 50% power to detect 100% of the original effect size (minimum threshold sample size).

For new data replications, power calculations were performed in accordance with the guidelines of the Social Sciences Replication Project[7]. The first round of data collection targeted the stage 1 sample size. If the original result was replicated according to the prespecified inferential criteria—the same pattern of association as the original finding with a two-tailed *P* value less than 0.05—no further data collection was carried out. If the original result did not replicate, a second round of data collection was added to the first, with the pooled sample targeting the stage 2 sample size. Note that one could object that we are informally performing sequential analyses using a 5% $\alpha$-level at each look at the data and thus inflating the Type 1 error rate. However:

1.  The false positive risk in this research is conservative because the replication test is (1) statistical significance and (2) directional consistency with the original finding. Our use of two-sided tests means that the nominal false positive risk is actually 2.5%, not 5%. Reference 7 simulation shows that the false positive risk at Stage 2, given this context, is 4.2%, still below the 5% threshold.
2.  Power estimates are somewhat conservative as they do not take into account the dependency of the Stage 1 and Stage 2 tests. (Reference 7 Supplementary Information pages 16.)

Nevertheless, this is an informal approach to sequential testing and could be improved in future research by adopting formal sequential testing methods instead[40]. Secondary data replications were only required to meet the minimum threshold sample size, although analysts were encouraged to select data sources that would result in the largest possible sample size.

**Preregistration and review.** All researcher teams conducting replications were required to preregister their study design and inferential criteria before implementation, using templates that mirrored the OSF preregistration template[44] supplemented with specific instructions for the SCORE programme. Once a draft preregistration was complete, a shared version was reviewed by between one and three reviewers who had expertise in health-related social-behavioural sciences before the researchers proceeded with their study (Editorial and Review Boards are available at https://www.cos.io/our-services/research/score/editorial-board/).

Reviewers examined the preregistered research designs for clarity, completeness and quality, and did not evaluate the quality of the original research, plausibility of the findings, or advisability of the design decisions that are appropriately faithful to the original methodology and findings. For all project types, the review period was set to 5 days, after which access to the document was closed to external participants. The research team was asked to address all outstanding comments from the reviewer and editor as soon as possible. A member of the Editorial Board oversaw the review of each paper and resolved any conflicts regarding design that were not resolved during the review process.

For new data replications, original authors were invited to participate in the review process by making comments and suggestions in the preregistration. After a preregistration was approved by the editor, original authors were invited to submit additional feedback as commentary. For secondary data replications, the original author was provided with a link to the finalized preregistration and invited to submit a commentary after the review period had closed. Since data had already been collected for the secondary data replications before preregistration, the original author was not invited to participate in the review to prevent undue influence in the development of the replication protocol. When provided, original author commentary was uploaded to the replication OSF project alongside the finalized preregistration. After registration, researchers proceeded with the full study implementation and returned outcome reports, summarized in Table 1. Details of all replication details, such as preregistrations, final outcome reports, materials used, power analysis, data and analysis, are included in Supplementary Information 1.

Importantly, despite these additional processes to review preregistrations and check for completeness of reporting, these individual replications did not go through peer review at a journal. The documentation of each replication is available on OSF and provides detailed information about each study, including any uncertainties noted by the researchers, although there might be additional unknown or unrecognized issues, which means there might be substantive issues that emerged while designing and executing the studies that might be open to criticism. Similar to other large-scale replication projects, the content on OSF will be maintained and updated if necessary, to ensure accuracy and transparency of the individual replication studies.

### Judgement elicitation for structured groups

**Participants.** We recruited 99 experienced participants from a pool of people who had previously participated in at least one repliCATS workshop or remote process for evaluating research claims, as described in ref. 33. One participant was an undergraduate student and was excluded from the subsequent analyses (another participant who was registered for an undergraduate degree had already completed another degree and was included in the experienced sample). The final experienced sample consists of 98 participants. Each experienced participant was awarded a US$200 grant to assess ten research claims for this study.

The 96 beginners were recruited from three undergraduate courses (ranging from an introductory course to a capstone one) taught by A.M. at the University of North Carolina, Chapel Hill. Eighty-eight beginners completed their assessments; only their data were included in the analysis. They were offered three extra credits in their respective courses to assess the replicability of 10–13 claims.

**Materials.** Using the methodology described above, 100 research claims were extracted from COVID-related social and behavioural science preprints, including 35 claims with known replication outcomes. We provide the bibliographic details for the preprints where claims were extracted from, in Supplementary Information 5.

**Procedures for elicitation of replication outcomes.** The prediction project was run independently from the replication project, and the team eliciting judgements from beginners and experienced participants were blinded to the methodology and results presented above (other than being made aware of which 100 papers were selected for replication) until after data collection was completed.

We divided the beginner and experienced samples into groups of 4–6. Participants were asked to specify which disciplines they felt comfortable assessing claims in and were assigned to groups such that the majority of claims people assessed were from disciplines they selected. They were invited to complete a standard demographics survey and to take a short quiz gauging their experience with open science practices and statistics, although answering questions was not enforced. Experienced participants each assessed ten claims (with each claim being assessed by two groups) and beginners assessed between 10–13 claims (with each claim being assessed by two groups). Data collection

took place from 28 August to 17 September 2020 for experienced participants and in two 3-week waves in the Fall 2020 and Spring 2021 semesters for beginners. All participants provided informed consent before commencing the study.

Participants evaluated their assigned claims on an online platform developed for the repliCATS project[33,45] (Fig. 2) implementing the 'IDEA' protocol[46–48] (Fig. 1). Before entering the platform for the first time, participants were redirected to a demographic survey and knowledge quiz. We collected information on participants' basic demographic characteristics (for example, age, gender) and professional experience. Participants were also asked to complete an optional knowledge quiz on statistical and metascience concepts (available on our OSF page, https://osf.io/4sfbj/). When participants logged on to the platform, they were assigned an avatar to obscure their identity. On the platform, the main page shows the list of papers assigned to that participant/account. Selecting a paper reveals a summary of the study containing the central claim made in the paper, key inferential statistics, the abstract and a link to the full paper. Participants then privately answered a series of questions about the paper, with the main question of interest being the replicability of the finding. Replication probability was defined as the likelihood that independent researchers repeating the study would find a statistically significant effect in the same direction as the original study (that is, Q3 in Table SM6, Supplementary Information 6). In this question, participants provided numerical judgements using a 3-point response format, first indicating the lower bound, then the upper bound and finally, their best estimate of the probability (0–100) that the claim would successfully replicate. This format encourages respondents to reflect on the underlying reasons for feasible high and low estimates and allows them to express their uncertainty by adjusting the interval width. Participants were also asked to describe, in a comment box on the platform, the reasoning behind their quantitative judgements.

After the first elicitation round, participants gained access to the Round 2 platform interface, which reveals the de-identified judgements and accompanying rationales provided by the other group members. Groups were then asked to discuss this information. Both experienced and beginner participants' discussions took place asynchronously, using the chat functionality implemented on the repliCATS platform (Fig. 2c). After a discussion, all participants were asked to review their judgements. They could view their original answers and answer the same key questions using the same format (Supplementary Information 6). This second round of judgements was private, that is, not visible to other group members. While they were not required to make any changes, they could do so if their assessment had changed following the discussion. Aggregate group judgements were then calculated according to the methods described below.

Both beginners and experienced participants were prompted to imagine that 'all replication studies have high power (90% power to detect an effect 50–75% of the original effect size with $\alpha = 0.05$, two-sided)'.

We note that we deviated slightly from our preregistered procedures in this process. The original preregistration was constrained by the goals and timelines of the DARPA-SCORE and hence, we expected only to be able to recruit undergraduate students in the Fall 2020 teaching semester at UNC. However, we were able to extend the timeline for this study and decided to run a second recruitment and elicitation round in the Spring 2021 teaching semester to secure roughly equal samples of 'experienced' and 'beginner' participants in the end. We further note an inconsistency in our preregistration. We stated that we will target recruiting 45 undergraduates in the 'Design plan' but only 40 in the sampling plan. We were indeed aiming for 45. Furthermore, the preregistration labels our two samples as novices ('beginners' in this paper) and experts ('experienced participants' in this paper). We decided to change the terminology used in the preregistration to clarify that we do not intend to suggest that any of the groups had in-depth

domain expertise in all the claims they assessed, but only that they were more experienced with assessing the quality (including replicability) of quantitative social and behavioural science results. We retain the term 'expertise' to explain the models used (Table 3), but this should be construed as 'task expertise'.

**Measures.** We used two accuracy measures in this study: error-based and classification accuracy. We calculated error-based accuracy as 1 minus the absolute difference between an estimate (0–1) and the replication outcome (0 for failed replications, 1 for successful replications). Classification accuracy was calculated as the proportion of estimates on the correct side of 0.5 (estimate ≥0.5 for successful replications, estimate <0.5 for failed replications) for a paper's replication outcome. Error-based accuracy was calculated at the individual and group level (that is, based on the pooled estimates within a given group). Classification accuracy was calculated at the individual and group levels but can only be modelled at the individual level and summarized at the group level.

**Analyses.** Our analyses were conducted using linear models or linear mixed-effects models and the individual models are outlined in Table 3. The models are grouped into two categories: behavioural models that were fit to the assessments of all 100 papers in the dataset, and accuracy models that were fit only to the subset of papers with a replication outcome and a replication power of ≥0.8 at a two-sided $\alpha$-level of 0.05 (29 papers) and a single-sided $\alpha$-level of 0.025 (24 papers, included as Supplementary Information 3). Behavioural models had response variables covering aspects of participant behaviour, such as confidence in their estimates (interval widths) and shifting behaviours between rounds. Accuracy models had response variables related to the participants' performance, including error-based and classification accuracy at the individual and group levels, and the shift in accuracy between rounds. Behavioural models included the paper ID as a random intercept, while accuracy models included replication type as random effects on the intercept. All models were fit in R (v.4.3.1)[49] using the 'lmerTest' R package (v.3.1-3)[50]. Degrees of freedom in the mixed-effects models were calculated using Satterthwaite's formula. We further examined our non-significant difference results using two one-sided tests[36] at an alpha level of 0.05 and an equivalence bound of ±0.05 raw mean difference scores using the 'TOSTER' R package (v.0.8.3)[51]. We report only the one-sided test that produces the higher $P$ value[36].

All correlations reported are Pearson correlations. The two experience variables, that is, 'previous experience with preregistration' and 'previous experience with metascience', are binary, and the remainder are continuous. To determine whether correlations between beginner vs experienced participants' judgements on the one hand and market scores on the other hand were significantly different, we specified a linear model lm($A \approx B + C + B \times C$), where $A$ = market scores, $B$ = participant judgements and $C$ = a dummy variable indicating the origin of the score as either beginner or experienced, and the interaction term ($B \times C$), to test for a significant difference between beginners and experienced participants.

The full model for the relationship between market scores and beginner versus experienced participants: $F(3, 196) = 27.08$, $P < 0.001$, adjusted $R^2 = 0.28$. Coefficients: intercept, $t(196) = 1.603$, $P = 0.111$, $\hat{\beta} = 0.105$, s.e. = 0.0654; participant scores, $t(196) = 8.031$, $P < 0.001$, $\hat{\beta} = 0.878$, s.e. = 0.1093; beginner/experienced, $t(196) = 1.686$, $P = 0.093$, $\hat{\beta} = 0.178$, s.e. = 0.1057; and the interaction term, $t(196) = -2.107$, $P = 0.036$, $\hat{\beta} = -0.354$, s.e. = 0.1682.

## Judgement elicitation for replication markets
In addition to the structured elicitation protocol, replication forecasts were elicited independently and simultaneously by a separate team ('the replication markets team') using a methodology established

in conjunction with earlier systematic replication projects[6,7,21,22]. The replication markets team used simple incentivized surveys and prediction markets to predict replication success. The methodology is outlined in detail in a preregistration plan (https://osf.io/t9xcn) and is summarized in ref. 34.

In the surveys, participants assessed the probability of a well-powered direct replication attempt to yield the same result as the original publication. Survey-based forecasting was done in batches of 10 claims. After completing their first batch, participants could provide forecasts for additional batches. They were incentivized by the surrogate scoring method[52] to provide accurate forecasts. This method uses predictions solicited from the 'peer' participants to compute an unbiased estimate for the accuracy of participant forecasts. Since surrogate scoring does not require a 'ground truth' signal to close the payments, payouts can be made once all forecasts are collected. For each batch, participants were ranked according to their expected accuracy and rewarded depending on their rank with fixed prices of US$80, 40, 20 and 20 for the four top-ranked forecasters. The ranking also allowed us to optimize the aggregation of survey results by allowing only forecasts with high expected accuracy to enter.

In the prediction market part, participants traded contracts with payoffs that depended on the outcome of a replication. Since only a subset of publications were expected to be assessed through replications, we scaled up the payoffs for contracts that were assessed through replications and thus could be resolved, and we voided the contracts that could not be resolved. Such an approach aligns with so-called decision markets[53] and allows the retention of incentive compatibility of prediction markets despite a high fraction of claims that cannot be resolved.

Most of the participants in the surveys and prediction markets participated in previous rounds of the SCORE forecasting project on published research from the social and behavioural sciences and were familiar with the forecasting tasks. In terms of experience, the cohort of participants resembled the experienced cohort rather than the beginner cohort in the structured elicitation protocol. Most participants were from academia and were graduate students or early-career researchers. On average, we had 37 participants per claim in the surveys and 16 participants per claim in the prediction markets.

In contrast to previous prediction markets, initial market prices were informed by the $P$ value associated with the focal claim in the original publication. Each claim was assigned into one of the following three categories ('$P \leq 0.001$', '$0.001 < P \leq 0.01$', '$P > 0.01$'), and the starting prices for these three categories were set to 0.8, 0.4 and 0.3, respectively.

### Ethics review
**Replications.** The procedures were approved by the local ethics review board at each institution that conducted replications with concurrence from the United States Army Medical Research and Development Command's Office of Research Protections, Human Research Protection Office (HRPO) or the United States Naval Information Warfare Center Pacific, HRPO.

**Judgement elicitation (structured groups).** The procedures were approved by the University of Melbourne (#1853445.6) and the University of North Carolina, Chapel Hill's Office of Human Research Ethics (#19-3104).

**Judgement elicitation (replication markets).** The procedures were approved by the Harvard University CUHS (#18-1729).

### Reporting summary
Further information on research design is available in the Nature Portfolio Reporting Summary linked to this article.

## Data availability
This study was preregistered at https://osf.io/5kcny. The full datasets of claim content, metadata for the 100 COVID-19 preprints, and the full dataset of all replication outcomes, including preregistrations and OSF projects, are available at https://doi.org/10.17605/OSF.IO/FJKSB ref. 54. The judgement elicitation datasets are available for the structured groups at https://osf.io/4sfbj/ ref. 55 and for the replication markets at http://osf.io/5kfc6/ ref. 56.

## Code availability
All scripts are available for the structured groups at https://osf.io/4sfbj/ ref. 55 and for the replication markets at http://osf.io/5kfc6/ ref. 56.

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

## Acknowledgements

This research was developed with funding from the Defense Advanced Research Projects Agency (DARPA) under cooperative agreement nos. HR001118S0047 (A.M., H.F., A.H., A.V., F.M., B.C.W., F.F., D.P.W.), HR00112020015 and N660011924015 (A.L.A., M.K.S., N.F., O.M., Z.L., E.S.P., A.H.T., B.L., T.M.E., B.A.N.) and N66001-19-C-4014 (T.P., F.H., M.J., Y.L.). Following the DARPA model, the funder played a role in conceptualization and design, but was not involved in data collection, analysis, decision to publish or preparation of the manuscript. The views, opinions and/or findings expressed are those of the authors and should not be interpreted as representing the official views or policies of the Department of Defense or the US Government. T.M.E., N.F., Z.L., B.L., O.M., B.N. and A.H.T. thank M. v. Assen, R. v. Aert, M. Bakker and M. Sitnikov for help with power calculations and calculation of effect sizes; V. Ashok for contributions to the replication outcome data; and the hundreds of researchers who conducted the replications, served as editors and reviewers for the preregistration review process, and helped conduct power analyses and additional statistical consulting. A.M. thanks D. Siegel for research assistance, and the audience of the SKAPE seminar at the University of Edinburgh for helpful feedback on an earlier version of this paper.

## Author contributions

For judgement elicitation (structured groups) and the overall manuscript: A.M. and H.F. performed conceptualization; D.P.W. curated data; D.P.W. and A.H. conducted formal analysis; F.F. acquired funding; A.M., H.F., D.P.W. and F.M. conducted investigations; A.M., H.F., D.P.W. and A.H. developed the methodology; A.M. administered the project; D.P.W. performed visualization; A.M., A.V., B.C.W., A.H. and D.P.W. wrote the original draft; A.M., A.V., B.C.W., A.H., D.P.W.

and F.F. reviewed and edited the manuscript. For claim extraction and replication management: T.M.E. and B.A.N. performed conceptualization; N.F., O.M., Z.L., A.L.A., E.S.P. and M.K.S. curated data; T.M.E. and B.A.N. acquired funding; N.F., A.H.T. and M.K.S. conducted investigations; N.F., O.M., A.H.T., M.K.S. and A.L.A. developed the methodology; N.F., O.M., Z.L., A.H.T., B.L., M.K.S., E.S.P. and A.L.A. administered the project; T.M.E. and B.A.N. supervised the project; O.M. and A.L.A. performed validation; O.M. and A.H.T. performed visualization; N.F., O.M., A.H.T. and T.M.E. wrote the original draft; O.M., A.H.T., B.L., T.M.E. and B.A.N. reviewed and edited the manuscript. For replications: S.v.d.L., J.R., K.U., N.H.-K., W.R.R., S.H., T.C., J.D., A.M.J., G.C., T.C., S.C., E.B., M.K., H.K., A.T., C.R.S., S.D., J.K., C.N., H.B., J.G.F., E.M.B., A.R.S., R.H. and Z.S. conducted investigations; S.v.d.L., J.R., K.U., N.H.-K., W.R.R., S.H., T.C., J.D., A.M.J., G.C., T.C., S.C., E.B., E.M., M.K., J.E.E., A.T., C.R.S., S.D., J.K., C.N., A.L.A., H.B., K.J.G.C., J.G.F., E.M.B., A.F. and Z.S. conducted formal analysis; E.B. and A.L.A. developed the methodology; N.H.-K., E.B., C.R.S., S.D., J.K., M.T., A.S. and A.L.A. curated the data; A.M.J., A.L.J.F. and A.L.A. administered the project. For judgement elicitation (replication markets): T.P. curated the data; T.P. and Y.L. wrote the original draft; T.P., Y.L., F.H. and M.J. reviewed and edited the manuscript.

## Competing interests

A.M. is a UKRI Policy Fellow seconded to the Department for Science, Innovation and Technology. The views and conclusions contained herein are those of the authors and should not be interpreted as representing the official policies, either expressed or implied, of the Department for Science, Innovation and Technology or the UK Government. B.A.N., T.M.E., O.M., Z.L., A.H.T., B.L., N.F., E.S.P., M.K.S. and A.L.A. are or were employees of the nonprofit Center for Open Science that has a mission to increase openness, integrity and reproducibility of research. The remaining authors declare no competing interests.

## Additional information

**Correspondence and requests for materials** should be addressed to Alexandru Marcoci.

Alexandru Marcoci [1,2,46] ✉, David P. Wilkinson [3,4,46], Ans Vercammen[3,5,6], Bonnie C. Wintle [3], Anna Lou Abatayo [7], Ernest Baskin [8], Henk Berkman[9], Erin M. Buchanan[10], Sara Capitán[11], Tabaré Capitán[12], Ginny Chan [13], Kent Jason G. Cheng[14], Tom Coupé[15], Sarah Dryhurst[16,17,18], Jianhua Duan[19], John E. Edlund[20], Timothy M. Errington [21], Anna Fedor [22], Fiona Fidler [3], James G. Field [23], Nicholas Fox[21], Hannah Fraser [3], Alexandra L. J. Freeman [17], Anca Hanea [3,24], Felix Holzmeister [25], Sanghyun Hong [15], Raquel Huggins[10], Nick Huntington-Klein[26], Magnus Johannesson [27], Angela M. Jones [28], Hansika Kapoor [29,30], John Kerr[17,31], Melissa Kline Struhl[32], Marta Kołczyńska[33], Yang Liu [34], Zachary Loomas [21], Brianna Luis [21], Esteban Méndez[35], Olivia Miske[21], Fallon Mody[3,36], Carolin Nast[37], Brian A. Nosek[21,38], E. Simon Parsons[21], Thomas Pfeiffer [39], W. Robert Reed [15], Jon Roozenbeek [16], Alexa R. Schlyfestone[10], Claudia R. Schneider[16,17,40], Andrew Soh[41], Zhongchen Song[42], Anirudh Tagat [43], Melba Tutor [44], Andrew H. Tyner[21], Karolina Urbanska [45] & Sander van der Linden [16]

[1]Centre for the Study of Existential Risk, University of Cambridge, Cambridge, UK. [2]School of Politics and International Relations, University of Nottingham, Nottingham, UK. [3]MetaMelb Research Initiative, University of Melbourne, Melbourne, Victoria, Australia. [4]QAECO, University of Melbourne, Melbourne, Victoria, Australia. [5]School of Communication and Arts, The University of Queensland, Brisbane, Queensland, Australia. [6]School of Population Health, Curtin University, Bentley, Western Australia, Australia. [7]Environmental Economics and Natural Resources Group, Wageningen University and Research, Wageningen, the Netherlands. [8]Department of Food, Pharma and Healthcare, Saint Joseph's University, Philadelphia, PA, USA. [9]Business School, University of Auckland, Auckland, New Zealand. [10]Analytics, Harrisburg University of Science and Technology, Harrisburg, PA, USA. [11]Department of Ecology, Swedish University of Agricultural Sciences, Uppsala, Sweden. [12]Department of Economics, Swedish University of Agricultural Sciences, Uppsala, Sweden. [13]Rhizom Psychological Services LLC, Atlanta, GA, USA. [14]Center for Healthy Aging, The Pennsylvania State University, University Park, PA, USA. [15]UCMeta, University of Canterbury, Christchurch, New Zealand. [16]Department of Psychology, University of Cambridge, Cambridge, UK. [17]Winton Centre for Risk and Evidence Communication, Department of Pure Mathematics and Mathematical Statistics, University of Cambridge, Cambridge, UK. [18]UCL Institute for Risk and Disaster Reduction, University College London, London, UK. [19]Statistics New Zealand, Christchurch, New Zealand. [20]Rochester Institute of Technology, Rochester, NY, USA. [21]Center for Open Science, Charlottesville, VA, USA. [22]Independent researcher, Budapest, Hungary. [23]Department of Management, John Chambers School of Business and Economics, West Virginia University, Morgantown, WV, USA. [24]Centre of Excellence for Biosecurity Risk Analysis, University of Melbourne, Melbourne, Victoria, Australia. [25]Department of Economics, University of Innsbruck, Innsbruck, Austria. [26]Seattle University, Seattle, WA, USA. [27]Department of Economics, Stockholm School of Economics, Stockholm, Sweden. [28]School of Criminal Justice and Criminology, Texas State University, San Marcos, TX, USA. [29]Department of Psychology, Monk Prayogshala, Mumbai,

India. [30]Neag School of Education, University of Connecticut, Storrs, USA. [31]Department of Public Health, University of Otago, Wellington, New Zealand. [32]Massachusetts Institute of Technology, Cambridge, MA, USA. [33]Institute of Political Studies, Polish Academy of Sciences, Warszawa, Poland. [34]Department of Computer Science and Engineering, University of California, Santa Cruz, Santa Cruz, CA, USA. [35]Central Bank of Costa Rica, San José, Costa Rica. [36]History and Philosophy of Science, University of Melbourne, Melbourne, Victoria, Australia. [37]University of Stavanger, School of Business and Law, Stavanger, Norway. [38]Department of Psychology, University of Virginia, Charlottesville, VA, USA. [39]NZ IAS, Massey University, Auckland, New Zealand. [40]School of Psychology, Speech and Hearing, University of Canterbury, Christchurch, New Zealand. [41]Department of Philosophy, University of Hawaii at Manoa, Honolulu, HI, USA. [42]New Zealand Institute of Economic Research (NZIER), Wellington, New Zealand. [43]Department of Economics, Monk Prayogshala, Mumbai, India. [44]Independent researcher, Quezon City, Philippines. [45]Independent researcher, Sheffield, UK. [46]These authors contributed equally: Alexandru Marcoci, David P. Wilkinson. ✉e-mail: alexandru.marcoci@gmail.com

# Reporting Summary

## Statistics

For all statistical analyses, confirm that the following items are present in the figure legend, table legend, main text, or Methods section.

| n/a | Confirmed | |
|---|---|---|
| ☐ | ☒ | The exact sample size (*n*) for each experimental group/condition, given as a discrete number and unit of measurement |
| ☐ | ☒ | A statement on whether measurements were taken from distinct samples or whether the same sample was measured repeatedly |
| ☐ | ☒ | The statistical test(s) used AND whether they are one- or two-sided *Only common tests should be described solely by name; describe more complex techniques in the Methods section.* |
| ☐ | ☒ | A description of all covariates tested |
| ☐ | ☒ | A description of any assumptions or corrections, such as tests of normality and adjustment for multiple comparisons |
| ☐ | ☒ | A full description of the statistical parameters including central tendency (e.g. means) or other basic estimates (e.g. regression coefficient) AND variation (e.g. standard deviation) or associated estimates of uncertainty (e.g. confidence intervals) |
| ☐ | ☒ | For null hypothesis testing, the test statistic (e.g. *F*, *t*, *r*) with confidence intervals, effect sizes, degrees of freedom and *P* value noted *Give P values as exact values whenever suitable.* |
| ☒ | ☐ | For Bayesian analysis, information on the choice of priors and Markov chain Monte Carlo settings |
| ☐ | ☒ | For hierarchical and complex designs, identification of the appropriate level for tests and full reporting of outcomes |
| ☐ | ☒ | Estimates of effect sizes (e.g. Cohen's *d*, Pearson's *r*), indicating how they were calculated |

*Our web collection on statistics for biologists contains articles on many of the points above.*

## Software and code

Policy information about availability of computer code

| Data collection | All scripts are available for the structured groups (https://osf.io/4sfbj/) and for the replication markets (http://osf.io/5kfc6/). |
|---|---|
| Data analysis | All analyses were conducted using R v4.3.1 and the code is available at osf.io/4sfbj/ |

For manuscripts utilizing custom algorithms or software that are central to the research but not yet described in published literature, software must be made available to editors and reviewers. We strongly encourage code deposition in a community repository (e.g. GitHub). See the Nature Portfolio guidelines for submitting code & software for further information.

## Data

Policy information about availability of data

All manuscripts must include a data availability statement. This statement should provide the following information, where applicable:
- Accession codes, unique identifiers, or web links for publicly available datasets
- A description of any restrictions on data availability
- For clinical datasets or third party data, please ensure that the statement adheres to our policy

The full datasets of claim content, metadata for the 100 COVID-19 preprints, and the full dataset of all replication outcomes, including preregistrations and OSF projects, are available at https://doi.org/10.17605/OSF.IO/FJKSB.
The judgement elicitation datasets are available for the structured groups (https://osf.io/4sfbj/) and for the replication markets (http://osf.io/5kfc6/).

# Research involving human participants, their data, or biological material

Policy information about studies with <u>human participants or human data</u>. See also policy information about <u>sex, gender (identity/presentation), and sexual orientation</u> and <u>race, ethnicity and racism</u>.

| | |
|---|---|
| Reporting on sex and gender | Information on repliCATS participant gender was collected using a survey, but this field was optional. Gender was not analysed directly and only used to show demographic characteristics of the expert and novice groups. Individual-level data on gender is not made publicly available. |
| Reporting on race, ethnicity, or other socially relevant groupings | We do not report on race, ethnicity or any other similar groupings. |
| Population characteristics | repliCATS collected demographic variables of participants (including age, education level) but they were not directly analysed and only used to show demographics characteristics of the expert and novice groups. |
| Recruitment | In order to test the predictive ability of participants with varying levels of task expertise, we recruited:<br><br>- 99 experts from a pool of people who had previously participated in at least one repliCATS workshop or remote process for evaluating research claims, described in Fraser et al. (2023). Each expert participant was awarded a $200USD grant to assess 10 research claims for this study.<br><br>- 96 novices from three undergraduate courses (ranging from an introductory course to a capstone one) taught by AM at the University of North Carolina, Chapel Hill. Eighty-eight novices completed their assessments and only their data has been included in the analyses. Novice participants were offered 3 extra credits in their respective courses to assess the replicability of 10-13 claims.<br><br>The sample is not representative. |
| Ethics oversight | Replications. The procedures were approved by the local ethics review board at each institution that conducted replications with concurrence from the United States Army Medical Research and Development Command's Office of Research Protections, Human Research Protection Office (HRPO) or the United States Naval Information Warfare Center Pacific, HRPO.<br><br>Judgement elicitation (structured groups). The procedures were approved by the University of Melbourne [#1853445.6] and the University of North Carolina, Chapel Hill's Office of Human Research Ethics [#19-3104].<br><br>Judgement elicitation (replication markets). The procedures were approved by the Harvard University CUHS [#18-1729]. |

Note that full information on the approval of the study protocol must also be provided in the manuscript.

# Field-specific reporting

Please select the one below that is the best fit for your research. If you are not sure, read the appropriate sections before making your selection.

☐ Life sciences        ☒ Behavioural & social sciences        ☐ Ecological, evolutionary & environmental sciences

For a reference copy of the document with all sections, see <u>nature.com/documents/nr-reporting-summary-flat.pdf</u>

# Behavioural & social sciences study design

All studies must disclose on these points even when the disclosure is negative.

| | |
|---|---|
| Study description | All analyses are quantitative and fit with either linear mixed effects models or simple correlations. There are some basic qualitative descriptors of the demographic characteristics of the participant pool. |
| Research sample | We recruited 99 experts from a pool of people who had previously participated in at least one repliCATS workshop or remote process for evaluating research claims, described in Fraser et al. (2023). Each expert participant was awarded a $200USD grant to assess 10 research claims for this study.<br><br>The 96 novices were recruited from three undergraduate courses (ranging from an introductory course to a capstone one) taught by AM at the University of North Carolina, Chapel Hill. Eighty- eight novices completed their assessments and only their data has been included in the analysis. Novice participants were offered 3 extra credits in their respective courses to assess the replicability of 10-13 claims. |
| Sampling strategy | The study sample was chosen such that we achieved the minimum number of assessments required per claim assessed from each group without providing too much of a cognitive burden on any given participant. The expert and novice group population samples are representative of people with and without experience assessing the replicability of scientific evidence. |
| Data collection | Expert elicitation data was collected using a custom-built online platform (described in Pearson et al. 2021). There was not anyone |

| Data collection | present besides the participants and members of our team who were facilitating the discussion. The researchers were not blind to the experimental condition or preregistered hypotheses during data collection. |
|---|---|
| Timing | Data collection took place from 28th August to 17th September 2020 for experts (experienced participants), and in two 3-week waves, in the Fall 2020 and Spring 2021 semesters for novices (beginner participants). |
| Data exclusions | We excluded data collected from one participant in the expert group as their highest level of completed education was high school. This was a pre-registered exclusion criteria. We also excluded 6 research claims from the results reported in the Main Text following reviewer 3's comments about low power. This was not a pre-registered exclusion criterion, so we include all results reported in the main text on the full dataset of 35 research claims as supplementary materials. Following a second round of reviews and subsequent conversations about how power should be computed we decided to include in the Discussion section and as an additional appendix a report of all our results from the Main Text for a sub-corpus 24 research claims (which excludes a further 5 claims). These exclusions are transparent and justified in the text. |
| Non-participation | Eight participants from the novice group signed up to participate but did not complete their assigned assessments and did not contribute any data used in the analysis. |
| Randomization | Participants were assigned to expert (experienced participants) or novice (beginner participants) groups based on their prior experience assessing the replicability of scientific evidence. Within these experimental groups participants were randomly allocated into smaller groups of 4-6 participants to assess any given piece of evidence. |

# Reporting for specific materials, systems and methods

We require information from authors about some types of materials, experimental systems and methods used in many studies. Here, indicate whether each material, system or method listed is relevant to your study. If you are not sure if a list item applies to your research, read the appropriate section before selecting a response.

## Materials & experimental systems

| n/a | Involved in the study |
|---|---|
| ☒ ☐ | Antibodies |
| ☒ ☐ | Eukaryotic cell lines |
| ☒ ☐ | Palaeontology and archaeology |
| ☒ ☐ | Animals and other organisms |
| ☒ ☐ | Clinical data |
| ☒ ☐ | Dual use research of concern |
| ☒ ☐ | Plants |

## Methods

| n/a | Involved in the study |
|---|---|
| ☒ ☐ | ChIP-seq |
| ☒ ☐ | Flow cytometry |
| ☒ ☐ | MRI-based neuroimaging |

## Plants

| Seed stocks | *Report on the source of all seed stocks or other plant material used. If applicable, state the seed stock centre and catalogue number. If plant specimens were collected from the field, describe the collection location, date and sampling procedures.* |
|---|---|
| Novel plant genotypes | *Describe the methods by which all novel plant genotypes were produced. This includes those generated by transgenic approaches, gene editing, chemical/radiation-based mutagenesis and hybridization. For transgenic lines, describe the transformation method, the number of independent lines analyzed and the generation upon which experiments were performed. For gene-edited lines, describe the editor used, the endogenous sequence targeted for editing, the targeting guide RNA sequence (if applicable) and how the editor was applied.* |
| Authentication | *Describe any authentication procedures for each seed stock used or novel genotype generated. Describe any experiments used to assess the effect of a mutation and, where applicable, how potential secondary effects (e.g. second site T-DNA insertions, mosiacism, off-target gene editing) were examined.* |

