## [Peer Review File · Nature Human Behaviour]

Peer Review Information

Journal: Nature Human Behaviour

Manuscript Title: Predicting the replicability of social and behavioural science claims in COVID-19 preprints

Corresponding author name(s): Alexandru Marcoci

Reviewer Comments & Decisions:

Decision Letter, initial version:

11th April 2023

Dear Dr Marcoci,

Thank you once again for your manuscript, entitled "Predicting the replicability of social and behavioural science claims from the COVID-19 Preprint Replication Project with expert and novice groups", and for your patience during the peer review process.

Your Article has now been evaluated by 3 referees. You will see from their comments copied below that, although they find your work of considerable potential interest, they have raised quite substantial concerns. In light of these comments, we cannot accept the manuscript for publication, but would be interested in considering a revised version if you are willing and able to fully address reviewer and editorial concerns.

We hope you will find the referees' comments useful as you decide how to proceed. If you wish to submit a substantially revised manuscript, please bear in mind that we will be reluctant to approach the referees again in the absence of major revisions. We are committed to providing a fair and constructive peer-review process. Do not hesitate to contact us if there are specific requests from the reviewers that you believe are technically impossible or unlikely to yield a meaningful outcome.

To guide the scope of the revisions, the editors discuss the referee reports in detail within the team, including with the chief editor, with a view to (1) identifying key priorities that should be addressed in revision and (2) overruling referee requests that are deemed beyond the scope of the current study. We hope that you will find the prioritised set of referee points to be useful when revising your study. Please do not hesitate to get in touch if you would like to discuss these issues further.

1. Reviewer 3 raises important concerns about the inflation of Type I error rates due to sequential analyses, as well as low power. We shared these concerns with the other reviewers, who agreed that this is an issue that should be addressed. Based on the additional feedback provided by our reviewers

in this respect (especially by Reviewer 1), we ask that you carry out a Bayesian analysis to assess the extent to which the collected data are diagnostic and can discriminate evidence of absence from absence of evidence. As Reviewer 1 noted to us, a Bayesian analysis would obviate the issue of power calculations and does not require sequential corrections.

2. Our reviewers ask that you more clearly explain the preprint selection process and motivate why you decided to replicate preprints specifically, instead of peer-reviewed publications. Please also follow the reviewers' advice and carefully check whether there were any deviations from preregistrations in the studies that you attempted to replicate.

3. In their reports, our reviewers raise several concerns about the conceptualization and definition of 'experts' and 'novices'. We ask that you thoroughly engage with these concerns and provide a clear definition of the terms 'experts' and 'novices'. Please also discuss the limitations and implications of these definitions for your findings.

4. Finally, please ensure that all replication studies are fully reported in Supplementary Information with a level of detail that would enable their reproduction.

If you wish to submit a suitably revised manuscript, we would hope to receive it within 4 months. I would be grateful if you could contact us as soon as possible if you foresee difficulties with meeting this target resubmission date.

- Include a "Response to the editors and reviewers" document detailing, point-by-point, how you addressed each editor and referee comment. If no action was taken to address a point, you must provide a compelling argument. When formatting this document, please respond to each reviewer comment individually, including the full text of the reviewer comment verbatim followed by your response to the individual point. This response will be used by the editors to evaluate your revision and sent back to the reviewers along with the revised manuscript.
- Highlight all changes made to your manuscript or provide us with a version that tracks changes.

[REDACTED]

Thank you for the opportunity to review your work. Please do not hesitate to contact me if you have

any questions or would like to discuss the required revisions further.

Sincerely,

[REDACTED]

Reviewer expertise:

Reviewer #1: replication ; metascience ; prediction markets

Reviewer #2: metascience

Reviewer #3: replication ; metascience

REVIEWER COMMENTS:

Reviewer #1:

Remarks to the Author:

This is a topical, well-executed, high-effort study that reports the results of 35 preregistered replications of COVID preprints. The assessment of forecasting ability itself was also preregistered. Although I believe the work ought to be published, I have several suggestions for further improvement.

Major comments

1. "We found that experts were strongly correlated ($r=.68$) to the final market prices, whereas novices were moderately correlated ($r=.35$, Figure 4)." But is this difference significant? The authors are interpreting an interaction, and this ought to be tested directly.
2. One of the main take-away points of this work may be that experts are remarkably unsure of whether or not a reported finding will actually replicate (!).
3. The abstract states that "expertise may not be required for credibility assessment of some research findings". It is true that novices and experts performed similarly; however, it is also true that their performance did not differ from chance. So it seems that "expertise does not help lift forecasting performance above chance levels" is a more apt summary of what the data show.

Minor comment

Could the authors clarify what estimates are shown in Figure 2? The authors say that these show "best estimates for each of the 35 known-outcome research claims" – but best estimates of what quantity? Is it the probability of a successful replication? Furthermore, the word "best" in "best estimates", suggests that in round two there were multiple estimates made, and the best was selected – what I believe is that "best estimates" refers to a participant's point estimate (?).

In general I recommend that the authors expand the figure headings so that it is unambiguously clear that a generic term such as "estimate" refers to.

I always sign my reviews,
E.J. Wagenmakers

Reviewer #2:

Remarks to the Author:

Marcoci et al. report the results of an important investigation into the replicability of social and behavioural science claims from COVID-19 preprints. The bottom line finding is that experts and novices are far from perfect at predicting replicability of claims, which has implications for research evaluation processes and methods for determining what to replicate in future. The paper is very well reported in general, but I had a few queries, which I outline below.

Introduction: Overall this was a really clear summary of prior literature and led to a strong rationale for the current study. However, there were a couple of places where I think more clarification is needed. In particular, as someone who has no idea what prediction markets are, I struggled to follow the sentence, "With a dichotomous prediction, markets achieved 73% accuracy (i.e., in 73% of cases the final market price for the contracts traded was $>.5$ for results that replicated and $<.5$ for results that did not) and prediction market prices correlated with replication outcomes ($r=.581$, $p<.001$, Gordon et al., 2021)" (line 128-131). Furthermore, I did not understand the "description-plus-evidence" scenario described in relation to the study by Hoogeveen et al. (line 159-165).

Introduction: The claim in the aims paragraph that, "Second, we elicited predictions about the replicability of claims published in preprints rather than journal articles" somewhat comes out of nowhere. I suggest the authors consider removing, or provide some text earlier raising the issue of replicability of preprints versus published studies.

The Results are presented before the Methods section. I realise this is a policy of the journal that the authors cannot change, but make a habit of pointing out that such a policy downplays the importance of study methods and makes reading the paper an unpleasant experience (not your fault, study authors!).

Methods – Selecting preprints: It is not clear how many assessors evaluated the preprints against the eligibility criteria for the study. If more than one assessor screened preprints, was this done independently, and if so, how were discrepancies in judgements resolved? And was there a staged process to screening preprints (e.g. titles/abstracts first, followed by full texts of those that were considered potentially relevant)?

Methods – Selecting preprints: The eligibility criterion – "It contained a claim that could feasibly be replicated in a good faith attempt by November 2020" – is somewhat vague, and I wonder if more detail could be provided here. I also suggest you defined what you mean by "replicated" as, as the authors would well know, there is a so much variation in definitions of replication.

Methods – Extracting claims from preprints: This section is really lacking in detail. It doesn't help a

reader to know that "Claims were extracted largely following the same methodology used in the rest of the SCORE program" without at least briefly explaining what method was used in the SCORE program.

Methods – Judgement elicitation: Structured groups: It was not clear to me who was defined as an "expert" and who was defined as a "novice". I understand how each party was sourced, but not what characteristics each needed to have. Did "experts" need to have expertise in COVID-19, the social and behavioural sciences, eliciting judgements or all of the above? And what undergraduate degrees did "novices" have to be enrolled in?

Results – No comments on this section. The narrative descriptions are very clear and nicely supplement the detailed tables and figures.

Discussion: I was surprised to see no paragraph(s) outlining the strengths and weaknesses of the study, of which I can think of many.

Reviewer #3:

Remarks to the Author:

It is clear the team of authors has performed an impressive amount of work. In their paper, they perform 35 replication studies of preprints that are in some way related to COVID-19 (some by collection new data, some by repeating analyses on public data sources). They also examine how well replication rates can be predicted, by experts, and novices at the University of North Carolina, Chapel Hill. The conclusions are that around 60% of the preprints yielded similar results in a replication study, and that replication success was difficult to predict.

The manuscript contains multiple parts. The 35 replication studies are a very nice dataset for future meta-scientific questions, and help to identify possible Type 1 errors in COVID-19 research. The ability of "experts" and "novices" to predict replicability, which according to the title and abstract is deemed by the authors as a main contribution, did not strike me as such interesting data given how experts and novices were sampled. I also observed some problems in the analysis, some doubts on how far the preregistered plan was followed in individual studies, and a lack of interpretability of the secondary replication studies, and a mismatch with how participants were asked this claim would be tested. I am limiting my comments here to the most substantial issues I found - if a revision is submitted, I would expand on the comments here with more detailed observations.

1) The study offers no good test of the difference between experts and novices.

The terms 'experts' and 'novices' indicate a difference in 'expertise' (e.g., in the abstract). It remains somewhat vague what our experts have expertise in. It does not seem to be expertise in the actual field of research. The authors say other studies sometimes define expertise as "Expertise was typically equated with having (or at least studying towards) a PhD in a cognate field." If other papers use this as a definition, I would consider it inadequate. The authors remains almost completely silent about any mechanism through which these "experts" are supposed to be able to predict replicability. As base-rate probability of H1 being true is one of the main predictors (logically speaking), one would require the experts to have expertise in base-rates of the hypotheses tested. But, the authors seem to completely ignore such essential reflections on mechanisms, and simply use the term 'expert' for someone who has at least some research experience in a field that is possibly related (nothing is

stated about a direct relation in the description of the sample). I find this problematic.

The only description of “experts” in the paper is “We recruited 99 experts from a pool of people who had previously participated in at least one replicATS workshop or remote process for evaluating research claims, described in Fraser et al. (2023).”. The “experts” are a highly heterogeneous group of graduate students and some more senior people, who have at least some research experience (it largely remains unclear how much and in what). This is not expertise. When I search for the definition of expertise I get “expert skill or knowledge in a particular field.”. That seems a solid definition of expertise to me. The authors provide no indication that the “experts” actually have expert skill in the field of COVID-19 research that the replication studies were about.

To be actual experts, the ‘experts’ should know something about, to name a few studies, “emotional response to Covid-19 humor in the Italian population during the lockdown”, “Republicans Are More Willing to Get Back to Social Life Because They Are Less Scared of COVID-19.” and “Unequal impact of structural health determinants and comorbidity on COVID-19 severity and lethality in older Mexican adults”. I think there is not a single person in the world who would call themselves an expert on these 3 topics – let alone all 35 topics that were eventually replicated. This is important, because the main variable that one could use to predict replicability is good knowledge about the base-rate of H1 in this field. Predicting replicability of individual findings is difficult under the best circumstances (Miller & Schwarz, 2011: Miller, J., & Schwarz, W. (2011). Aggregate and individual replication probability within an explicit model of the research process. *Psychological Methods*, 16(3), 337–360. <https://doi.org/10.1037/a0023347>), and especially so because it is strongly based on the base rate probability that H1 is true. How are these “experts” supposed to have this knowledge? Which mechanisms do the authors expect constitute the causal relationship between having some research experience, and being able to predict replicability? As it is currently written up, this causal mechanism remains completely vague.

Then, when we come to the ‘novices’, there is a smaller but also important issue. The novices are “recruited from three undergraduate courses (ranging from an introductory course to a capstone one) taught by AM at the University of North Carolina, Chapel Hill.” Here, the problem is of course generalizability. I have no idea what this group consists of in terms of lack of expertise. It is not even stated which undergraduate courses they enrolled in. Did these students know anything about empirical research, social science, research methods? It remains completely unspecified – almost as if the authors do not really care about how they are manipulating being an expert vs. being a novice. But in any case, it is difficult to generalize from this sample to all ‘novices’. That limitation needs to be acknowledged.

Finally, the samples are not described in terms of where they groups come from. I understand it would look slightly awkward to have to state in a table that all novices come from the same 3 courses at a university in North America, but I am more interested in where the ‘experts’ come from. How many have expertise about the situation in Italy, or Mexico, or the US, given that the 3 studies I randomly picked above are about this context? It seems only normal to describe where participants are from – indeed Table M1 even summarizes the geographic context of the original studies.

So, in my view, the idea that this study tested anything about ‘experts’ predicting the replicability of COVID-19 studies does not fly. I think it is difficult to make a decent argument why expertise *as manipulated in this paper* would be a relevant factor to study. To me (although of course this is easy to say in hindsight) I do not see any real mechanism that could have led either group to perform better than the other predicting the replicability of these studies. A possible mechanism would have been that some studies have been p-hacked, and that people with research expertise have learned

about this and can identify it, but the students at the University of North Carolina, Chapel Hill had not been taught about p-hacking yet. However, even such mechanisms seem very far removed from the real research question the authors posit, which is not just if experts can make good predictions about replicability, but if eliciting predictions could “accelerate credibility assessment”. The fact that neither of these groups can predict replicability, given that (as mentioned above) replicability is difficult to predict even under the best circumstances, is not that useful a contribution.

If the authors believe the main contribution of this paper is the predictability by experts, and how likely it is that expert predictions play a role in assessing the credibility of findings, then I think the contribution is very low due to the lack of real expertise in the ‘experts’ and, to a lesser extent, by the lack of generalizability of the performance of the ‘novices’ who were all samples from 3 courses at one north-American university. This question could probably be improved by largely acknowledging that the expert vs novice distinction was not manipulated well, and that the study examines how people with no knowledge about the research topic, and with some research experience, make predictions about replicability.

2) Why COVID studies?

I have to start by admitting that from an extremely personal and subjective belief, I doubt a single COVID-19 study performed by social scientists was worth replicating. To me, the research interest in this topic seems to me to have been a huge research waste, as thousands of studies have been performed, but almost none of them seem to have been used for anything. Of course, I am happy to be proven wrong, and the authors could easily do so by pointing out why some of the 100 studies in this set of studies was important, either theoretically, or practically. However, the authors do not give any indication why it is worthwhile to replicate COVID-19 studies. The only justification of choosing this field I could find in the article is: “First, our forecasting questions were drawn from research in an emerging field, i.e. the COVID-19 pandemic”. I personally do not think the authors are correct to refer to COVID-19 studies as an ‘emerging field’ because the studies clearly fall in many different research fields, at least as these are traditionally defined. Preprints were even searched in 3 preprint servers related to completely different disciplines (medicine, psychology, sociology).

As explained in my 2 articles that the authors cite in their introduction, I believe we need to think carefully about which studies we replicate, given limited resources. In this case, the authors provide no reason whatsoever for why these COVID-19 studies were worthy of replication. Typical reasons in the literature to replicate studies is because they are a random sample of some population of interest, or because the findings in the papers were important. The first could be the case here, but it is not explained. The second, given that these are preprints that had not yet had an impact, is not the case. So, I am left wondering why the authors replicated these COVID-19 studies, and what their replications mean for their respective research areas. How many of these preprints were eventually published? Were papers that did not replicate published and cited a lot, or did they remain preprints and failed to pass peer review. I have the feeling an analysis of similar questions could start a reflection on why it was important to replicate this heterogenous set of studies, and if replicability is related in any way with publication/citation.

3. Sequential analysis and inadequate sample size justifications for secondary data analyses.

The authors used a peculiar data collection procedure, where they seem to informally perform sequential analyses. Quite problematically, they seem to be using a 5% alpha level at each look at the data. They cite a previous study that did this – but this is irrelevant, because a flaw is a flaw, even if it passed peer review previously. It should be obvious that looking at the data twice inflates the Type 1

error rate, and given that sequential analysis procedures exist that can control the Type 1 error, these should be used. I understand that might require a deviation from the preregistration plan for these studies. However, bad preregistrations should not be followed, so the deviation is necessary, as the study plan was incompetently designed. The authors might want to consider adding a methodologist or statistician to their team to prevent such mistakes in the future, and to choose the best way to correct this oversight now it has occurred.

For secondary data, the sample size justification that was followed was: "the sample size required to achieve 50% power to detect 100% of the original effect size (minimum threshold sample size)." This is completely inadequate. Studies that have 50% power lead to no better rate of claims if there is a true effect than a coin flip – and a coin flip is a lot cheaper. Maybe this is a typo – because the authors did not share sufficient information about the studies (e.g., a link to the OSF) I can only take the analyses as briefly summarized in the article. The secondary replication studies are therefore not informative.

For example, take the study by Simone & Gnagnarella, 2020. The replication yielded a $p = .214$. With 50% power, that is not unlikely even if there is a true effect. The replication of Seale et al., 2020 has a much smaller sample size as the original as well, and a non-significant result, as does Flesia et al., 2020. Messner & Payson, 2020 has a $p = 0.089$ in the replication.

4. Secondary data collection should be separated from novel data collection.

In general the secondary data replications are therefore peculiar. Conceptually, they are also a bit of a different beast than replication studies based on new data. I do not believe both types of studies should be combined in a single analysis. First, their Type 2 error rate was less controlled. But more importantly, the predictions were not made based on the idea that these replications would only have 50% power, as the instructions of the participants, if I understand it well enough, was: Note that both experts and novices were prompted to imagine that "all replication studies have high power (90% power to detect an effect 50-75% of the original effect size with $\alpha=0.05$, two-sided)". As this is not true for the secondary data replications, why did the authors believe that these studies could be included in the analyses of the prediction success? I think moving the secondary data collections to a supplement is best, given how uninformative they are, and how they can not be used in computing prediction success, as the predictors were asked to imagine the claims being tested with higher power.

5. Checking all replication studies for deviations from the preregistered plan

The authors seem to be releasing all replication studies from this COVID project alongside this article. It took me a bit too long to figure out that Table 1 has the links to individual replication studies. I think at least one reviewer should spent time checking all these studies.

I started by checking one study, and found something that surprised me. I wanted to check study osf.io/kbm46 because the original had 78 countries, the replication had 103 countries. This seemed surprisingly low to me, given the sampling plan.

In the files <https://osf.io/cvmzh> I find the text:

For these calculations, the primary unit of analysis are Country. An estimate of the minimum viable sample size for the data analytic replication is: 41. For comparison, the stage1 required sample size would be: 201 and the stage2 sample size would be: 454.

(Note, I want to give the author of this text credit for acknowledging "Confidence is low only because I am not deeply familiar with log linear." for this power analysis.). Nevertheless, it seems like this study

did not manage to collect the required data of 201 countries. The observed p-value was .14 in the replication. Was this a null due to low power?

I checked another study with a lower sample size in the replication than the original. osf.io/ytuk9 leads to <https://osf.io/926w4> which states:

For these calculations, the primary unit of analysis are participants. A preliminary estimate for the required sample size is **a sample of 243 for stage 1 data collection**, collecting an additional 303 to reach **a total of 546** if stage 2 data collection is conducted.

Despite a non-significant result after stage 1, with $p = .1203$, no second sample seems to have been collected, thus violating the sample size plan?

I wonder in how many studies there are unreported deviations from the plan described in the submitted manuscript, but these 2 checks make me believe someone should carefully check all studies.

If I read: <https://osf.io/5kcny> I see that the preregistration stated the number of novices as 45 and as 40 (so, the preregistration is not consistent). This sample size is not in line with the collected data – there were more novices, which is not a problem, but there is no explanation of why this is the case or how this happened, as the expectation was “The number of novice participants is constrained by the availability of students in the relevant classes at UNC”.

This link osf.io/t9xcn was not accessible, and should have been turned into a peer review only link before submission.

6. Minor comment:

In the abstract the statement “Replication is an important “credibility control” mechanism” for clarifying the reliability of published findings.” is vague. A direct replication (as performed here) helps to identify Type 1 errors. Less direct replications help to test generalizability across the dimensions that are varied. Terms such as ‘credibility’ and ‘reliability’ are unnecessarily vague – they can be used, but then they need to be explicitly linked to epistemological goals. Credibility could be linked to an increase in subjective belief, and reliability to tests of validity (such as generalizability) – but such links and definitions need to be explicit.

Use of qualifiers throughout the text. In the abstract, authors write “claims were strongly correlated ($r=.48$).” Why is this deemed a ‘strong’ correlation? It is a very low correlation if one would consider if the groups are equivalent (as there is clear variation between groups). What is it ‘strong’ for?

Symbols are not printed in the sentence: three categories (“ $_ \leq 0.001$ ”, “ $0.001 < _ \leq 0.01$ ”, “ $_ > 0.01$ ”),

Round 2 ($\beta = 0547$, > a comma must be missing somewhere.

Signed,

Daniel Lakens

Author Rebuttal to Initial comments

REVIEWER COMMENTS:

Reviewer #1:

Remarks to the Author:

This is a topical, well-executed, high-effort study that reports the results of 35 preregistered replications of COVID preprints. The assessment of forecasting ability itself was also preregistered. Although I believe the work ought to be published, I have several suggestions for further improvement.

Major comments

1. “We found that experts were strongly correlated ($r=.68$) to the final market prices, whereas novices were moderately correlated ($r=.35$, Figure 4).” But is this difference significant? The authors are interpreting an interaction, and this ought to be tested directly.

→ *We thank the reviewer for pushing us to further investigate this and we have now included a linear model with an interaction term to test if there was a difference in the correlations between expert and novice group scores with the Markets prices. We report this in the main text of the manuscript and explain the model used in the Methods section.*

2. One of the main take-away points of this work may be that experts are remarkably unsure of whether or not a reported finding will actually replicate (!).

→ *This is a good point, and we now include a slightly modified claim in the Discussion section. However, we now explicitly acknowledge the limitations to both our conceptualisation of expertise and our manipulation, and the fact that there is arguably greater uncertainty in this study than in other ‘forecasting replication’ studies, because it deals with ‘fast science’. Therefore, we are now more conservative in our conclusions.*

3. The abstract states that “expertise may not be required for credibility assessment of some research findings”. It is true that novices and experts performed similarly; however, it is also true that their performance did not differ from chance. So it seems that “expertise does not help lift forecasting performance above chance levels” is a more apt summary of what the data show.

→ *We have now revised both the results and their interpretation in response to Reviewer 3. In particular, we restricted all accuracy analyses to the subset of 28 high-powered replications (see the new Table 2). This restriction has led to both groups*

performing better than chance on most measures of accuracy we looked at. It remains the case that so-called “expertise” does not help lift forecasting performance above “novice” (now re-labelled as ‘beginners’) performance and it is generally lower than in other studies. However, this is most likely influenced by our choice of focusing on preprints about COVID-19, which leads to evaluations under ‘conditions of high uncertainty’. We now elaborate on these observations in both the introduction and discussion section.

Minor comment

Could the authors clarify what estimates are shown in Figure 2? The authors say that these show “best estimates for each of the 35 known-outcome research claims” – but best estimates of what quantity? Is it the probability of a successful replication? Furthermore, the word “best” in “best estimates”, suggests that in round two there were multiple estimates made, and the best was selected – what I believe is that “best estimates” refers to a participant’s point estimate (?).

In general I recommend that the authors expand the figure headings so that it is unambiguously clear that a generic term such as “estimate” refers to.

→ All participants provided estimates in two elicitation rounds using an interval answer format. That is, participants were prompted to provide their (i) lower bound, (ii) upper bound and (iii) best estimate of the probability a claim will replicate successfully (statistical significance and in the same direction as in the original paper). We now include this explanation in Results/section 3, and it is further developed in the Methods section. We decided to continue reporting ‘best estimates’ as this is the way the question was asked in all materials, but we hope the additional explanations included in the Results section will make this terminology more transparent to readers.

I always sign my reviews,
E.J. Wagenmakers

Reviewer #2:

Remarks to the Author:

Marcoci et al. report the results of an important investigation into the replicability of social and behavioural science claims from COVID-19 preprints. The bottom line finding is that experts and novices are far from perfect at predicting replicability of claims, which has implications for research evaluation processes and methods for determining what to replicate in future. The paper is very well reported in general, but I had a few queries, which I outline below.

Introduction: Overall this was a really clear summary of prior literature and led to a strong rationale for the current study. However, there were a couple of places where I think more clarification is needed. In particular, as someone who has no idea what prediction markets are, I struggled to follow the sentence, “With a dichotomous prediction, markets achieved 73% accuracy (i.e., in 73% of cases the final market price for the contracts traded was $>.5$ for results that replicated and $<.5$ for results that did not) and prediction market prices correlated with replication outcomes ($r=.581$, $p<.001$, Gordon et al., 2021)” (line 128-131). Furthermore, I did not understand the “description-plus-evidence” scenario described in relation to the study by Hoogeveen et al. (line 159-165).

→ *We have now added a brief plain language explanation of prediction markets and made the difference between “description-only” and “description-plus-evidence” in Hoogeveen et al. clearer.*

Introduction: The claim in the aims paragraph that, “Second, we elicited predictions about the replicability of claims published in preprints rather than journal articles” somewhat comes out of nowhere. I suggest the authors consider removing, or provide some text earlier raising the issue of replicability of preprints versus published studies.

→ *We thank the reviewer for this suggestion, and we have now included a new paragraph in the introduction better framing our choice of focusing on preprints. This paragraph also responds to some of the concerns raised by Reviewer 3. We then return to this issue in the discussion.*

The Results are presented before the Methods section. I realise this is a policy of the journal that the authors cannot change, but make a habit of pointing out that such a policy downplays the importance of study methods and makes reading the paper an unpleasant experience (not your fault, study authors!).

→ *Indeed, very little we can do about that. However, we have now included additional brief explanations of our methods in the results section to help readers better understand our findings. We also included more references to the Methods part of the paper to signal to the reader where important information required for interpreting our findings can be found.*

Methods – Selecting preprints: It is not clear how many assessors evaluated the preprints against the eligibility criteria for the study. If more than one assessor screened preprints, was this done independently, and if so, how were discrepancies in judgements resolved? And was there a staged process to screening preprints (e.g. titles/abstracts first, followed by full texts of those that were considered potentially relevant)?

→ *We thank the reviewer for prompting us to be more transparent about how we selected preprints. We included additional details regarding the selection of preprints in the methods section, responding to all these questions.*

Methods – Selecting preprints: The eligibility criterion – “It contained a claim that could feasibly be replicated in a good faith attempt by November 2020” – is somewhat vague, and I wonder if more detail could be provided here. I also suggest you defined what you mean by “replicated” as, as the authors would well know, there is a so much variation in definitions of replication.

→ *We include additional details regarding the eligibility criterion in the methods section.*

Methods – Extracting claims from preprints: This section is really lacking in detail. It doesn't help a reader to know that “Claims were extracted largely following the same methodology used in the rest of the SCORE program” without at least briefly explaining what method was used in the SCORE program.

→ *We include additional details about extracting claims from preprints in the methods section.*

Methods – Judgement elicitation: Structured groups: It was not clear to me who was defined as an “expert” and who was defined as a “novice”. I understand how each party was sourced, but not what characteristics each needed to have. Did “experts” need to have expertise in COVID-19, the social and behavioural sciences, eliciting judgements or all of the above? And what undergraduate degrees did “novices” have to be enrolled in?

→ *One of the major revisions we made regards how the difference between the two groups is construed, and what we can learn from how expertise was manipulated in this study. We also introduced the notion of ‘task expertise’ and distinguished it from ‘domain expertise’ in the COVID-19 pandemic. Our experienced participants had task expertise - experience with both quantitative methodology in the social and behavioural sciences and evaluating & critiquing social science research (as opposed to expertise in each individual paper, which covered a huge range of topics, as Reviewer 3 correctly points out, below). Additionally, we acknowledged some of the limitations stemming from our design choices in the discussion and reported on more demographics information we collected that emphasise the difference in task expertise between our two groups of participants. Some of the characteristics of our ‘experienced participants’ reported in Table 2 have been found to correlate with improved performance in forecasting the outcome of replications (Wintle et al., 2023)*

Results – No comments on this section. The narrative descriptions are very clear and nicely supplement the detailed tables and figures.

Discussion: I was surprised to see no paragraph(s) outlining the strengths and weaknesses of the study, of which I can think of many.

→ *We elaborate on the limitations of our study in the discussion section*

Reviewer #3:

Remarks to the Author:

It is clear the team of authors has performed an impressive amount of work. In their paper, they perform 35 replication studies of preprints that are in some way related to COVID-19 (some by collection new data, some by repeating analyses on public data sources). They also examine how well replication rates can be predicted, by experts, and novices at the University of North Carolina, Chapel Hill. The conclusions are that around 60% of the preprints yielded similar results in a replication study, and that replication success was difficult to predict.

The manuscript contains multiple parts. The 35 replication studies are a very nice dataset for future meta-scientific questions, and help to identify possible Type 1 errors in COVID-19 research. The ability of “experts” and “novices” to predict replicability, which according to the title and abstract is deemed by the authors as a main contribution, did not strike me as such interesting data given how experts and novices were sampled. I also observed some problems in the analysis, some doubts on how far the preregistered plan was followed in individual studies, and a lack of interpretability of the secondary replication studies, and a mismatch with how participants were asked this claim would be tested. I am limiting my comments here to the most substantial issues I found - if a revision is submitted, I would expand on the comments here with more detailed observations.

1) The study offers no good test of the difference between experts and novices.

The terms ‘experts’ and ‘novices’ indicate a difference in ‘expertise’ (e.g., in the abstract). It remains somewhat vague what our experts have expertise in. It does not seem to be expertise in the actual field of research. The authors say other studies sometimes define expertise as “Expertise was typically equated with having (or at least studying towards) a PhD in a cognate field.” If other papers use this as a definition, I would consider it inadequate. The authors remains almost completely silent about any mechanism through which these “experts” are supposed to be able to predict replicability. As base-rate probability of H1 being true is one of the main predictors (logically speaking), one would require the experts to have expertise in base-rates of the hypotheses tested. But, the authors seem to completely ignore such essential reflections on mechanisms, and simply use the term ‘expert’ for someone who has at

least some research experience in a field that is possibly related (nothing is stated about a direct relation in the description of the sample). I find this problematic.

The only description of “experts” in the paper is “We recruited 99 experts from a pool of people who had previously participated in at least one repliCATS workshop or remote process for evaluating research claims, described in Fraser et al. (2023).”. The “experts” are a highly heterogeneous group of graduate students and some more senior people, who have at least some research experience (it largely remains unclear how much and in what). This is not expertise. When I search for the definition of expertise I get “expert skill or knowledge in a particular field.”. That seems a solid definition of expertise to me. The authors provide no indication that the “experts” actually have expert skill in the field of COVID-19 research that the replication studies were about.

To be actual experts, the ‘experts’ should know something about, to name a few studies, “emotional response to Covid-19 humor in the Italian population during the lockdown”, “Republicans Are More Willing to Get Back to Social Life Because They Are Less Scared of COVID-19.” and “Unequal impact of structural health determinants and comorbidity on COVID-19 severity and lethality in older Mexican adults”. I think there is not a single person in the world who would call themselves an expert on these 3 topics – let alone all 35 topics that were eventually replicated. This is important, because the main variable that one could use to predict replicability is good knowledge about the base-rate of H1 in this field. Predicting replicability of individual findings is difficult under the best circumstances (Miller & Schwarz, 2011; Miller, J., & Schwarz, W. (2011). Aggregate and individual replication probability within an explicit model of the research process. *Psychological Methods*, 16(3), 337–360. <https://doi.org/10.1037/a0023347>), and especially so because it is strongly based on the base rate probability that H1 is true. How are these “experts” supposed to have this knowledge? Which mechanisms do the authors expect constitute the causal relationship between having some research experience, and being able to predict replicability? As it is currently written up, this causal mechanism remains completely vague.

Then, when we come to the ‘novices’, there is a smaller but also important issue. The novices are “recruited from three undergraduate courses (ranging from an introductory course to a capstone one) taught by AM at the University of North Carolina, Chapel Hill.” Here, the problem is of course generalizability. I have no idea what this group consists of in terms of lack of expertise. It is not even stated which undergraduate courses they enrolled in. Did these students know anything about empirical research, social science, research methods? It remains completely unspecified – almost as if the authors do not really care about how they are manipulating being an expert vs. being a novice. But in any case, it is difficult to generalize from this sample to all ‘novices’. That limitation needs to be acknowledged.

Finally, the samples are not described in terms of where they groups come from. I understand it would look slightly awkward to have to state in a table that all novices come from the same 3 courses at a university in North America, but I am more

interested in where the ‘experts’ come from. How many have expertise about the situation in Italy, or Mexico, or the US, given that the 3 studies I randomly picked above are about this context? It seems only normal to describe where participants are from – indeed Table M1 even summarizes the geographic context of the original studies.

So, in my view, the idea that this study tested anything about ‘experts’ predicting the replicability of COVID-19 studies does not fly. I think it is difficult to make a decent argument why expertise *as manipulated in this paper* would be a relevant factor to study. To me (although of course this is easy to say in hindsight) I do not see any real mechanism that could have led either group to perform better than the other predicting the replicability of these studies. A possible mechanism would have been that some studies have been p-hacked, and that people with research expertise have learned about this and can identify it, but the students at the University of North Carolina, Chapel Hill had not been taught about p-hacking yet. However, even such mechanisms seem very far removed from the real research question the authors posit, which is not just if experts can make good predictions about replicability, but if eliciting predictions could “accelerate credibility assessment”. The fact that neither of these groups can predict replicability, given that (as mentioned above) replicability is difficult to predict even under the best circumstances, is not that useful a contribution.

If the authors believe the main contribution of this paper is the predictability by experts, and how likely it is that expert predictions play a role in assessing the credibility of findings, then I think the contribution is very low due to the lack of real expertise in the ‘experts’ and, to a lesser extent, by the lack of generalizability of the performance of the ‘novices’ who were all samples from 3 courses at one north-American university. This question could probably be improved by largely acknowledging that the expert vs novice distinction was not manipulated well, and that the study examines how people with no knowledge about the research topic, and with some research experience, make predictions about replicability.

→ *We thank the reviewer for their careful reading of our manuscript and detailed analysis. We largely agree with these comments, and have revised the manuscript to:*

1) clarify that we are talking about groups with different levels of task-experience (i.e. experience relevant to the task of evaluating claims, which requires some expertise in assessing methodology, experimental design, statistical analysis, and critiquing scientific papers). We have clarified that we are not talking about domain expertise here. Even if our “experts” do have more domain expertise on evaluating some claims, we agree that they are unlikely to have relevant domain expertise across claims. However, the two groups are qualitatively different (undergrads vs postgrads / ECRs and more experienced researchers), which is reflected in our demographics data (Table 2). In particular, the “novices” have substantially less self-reported experience with metascience and preregistration of studies and did not perform as well on our quiz that tested their knowledge of statistical concepts. They also reported lower familiarity with

statistics, probability and experimental design and overall had taken fewer courses in quant methods and published fewer academic papers. That being said, our beginners had taken on average 1 course in quantitative methods and were given the option to choose which fields they wanted to assess papers from, in order to better match their experience.

2) we have revised the language around expertise, i.e., rather than calling our two groups “novices” and “experts”, we have called them “beginners” and “experienced” (i.e. in terms of task-relevant expertise). We agree that the term “experts” gives the impression that we are talking about domain expertise, which is misleading. We have also removed this from the title.

3) we have included several paragraphs that present some of the limitations Reviewer 3 and others have raised in the discussion, and we now also acknowledge the challenges to generalizability given how our participants were sampled. We also included more detailed information regarding our participants’ geographical location in Table 2.

4) finally, we have reframed the study and substantially changed the discussion to better motivate our manipulation of expertise and better explain the findings of this study.

2) Why COVID studies?

I have to start by admitting that from an extremely personal and subjective belief, I doubt a single COVID-19 study performed by social scientists was worth replicating. To me, the research interest in this topic seems to me to have been a huge research waste, as thousands of studies have been performed, but almost none of them seem to have been used for anything. Of course, I am happy to be proven wrong, and the authors could easily do so by pointing out why some of the 100 studies in this set of studies was important, either theoretically, or practically. However, the authors do not give any indication why it is worthwhile to replicate COVID-19 studies. The only justification of choosing this field I could find in the article is: “First, our forecasting questions were drawn from research in an emerging field, i.e. the COVID-19 pandemic”. I personally do not think the authors are correct to refer to COVID-19 studies as an ‘emerging field’ because the studies clearly fall in many different research fields, at least as these are traditionally defined. Preprints were even searched in 3 preprint servers related to completely different disciplines (medicine, psychology, sociology). As explained in my 2 articles that the authors cite in their introduction, I believe we need to think carefully about which studies we replicate, given limited resources. In this case, the authors provide no reason whatsoever for why these COVID-19 studies were worthy

of replication. Typical reasons in the literature to replicate studies is because they are a random sample of some population of interest, or because the findings in the papers were important. The first could be the case here, but it is not explained. The second, given that these are preprints that had not yet had an impact, is not the case.

So, I am left wondering why the authors replicated these COVID-19 studies, and what their replications mean for their respective research areas. How many of these preprints were eventually published? Were papers that did not replicate published and cited a lot, or did they remain preprints and failed to pass peer review. I have the feeling an analysis of similar questions could start a reflection on why it was important to replicate this heterogenous set of studies, and if replicability is related in any way with publication/citation.

→ *We largely agree with these comments, and we now provide additional details in the introduction and discussion sections motivating our choice of eliciting judgments about COVID-19 pandemic preprints. We also include an explicit limitation in the discussion regarding our sampling of preprints. The purpose of the study was to test the accuracy of human prediction about the replicability of COVID-19 research, and preprints were selected for this purpose. We did not intend to investigate the quality of COVID-19 research overall and we did not sample preprints to this end. Finally, this study was part of the DARPA-SCORE program which in March 2020 began including COVID-19 claims in the social and behavioural science, in response to the fast-developing global emergency. Finally, we decided not to provide information regarding which preprints ended up being published. We provide, however, full information about all preprints included in this study in the Methods section and readers can easily search for this information.*

3. Sequential analysis and inadequate sample size justifications for secondary data analyses.

The authors used a peculiar data collection procedure, where they seem to informally perform sequential analyses. Quite problematically, they seem to be using a 5% alpha level at each look at the data. They cite a previous study that did this – but this is irrelevant, because a flaw is a flaw, even if it passed peer review previously. It should be obvious that looking at the data twice inflates the Type 1 error rate, and given that sequential analysis procedures exist that can control the Type 1 error, these should be used. I understand that might require a deviation from the preregistration plan for these studies. However, bad preregistrations should not be followed, so the deviation is necessary, as the study plan was incompetently designed. The authors might want to consider adding a methodologist or statistician to their team to prevent such mistakes in the future, and to choose the best way to correct this oversight now it has occurred.

→ *The rationale for this procedure can be found in Camerer et al., 2018. To summarize the key points:*

[1] False positive risk in this research is conservative because the replication test is (a) statistical significance and (b) directional consistency with the original finding. Our use of two-sided tests means that the nominal false positive risk is actually 2.5% not 5%. The Camerer et al. simulation shows that the false positive risk at Stage 2 given this context is 4.2%, still below the 5% threshold.

[2] Power estimates are somewhat conservative as they do not take into account the dependency of the Stage 1 and Stage 2 tests.

Quote from Camerer et al., 2018 (Supplementary Information pages 16): “Note that if a study fails to replicate in the first data collection, it is given a second chance to replicate and two tests are conducted. This increases the false positive risk somewhat compared to carrying out a single test. However, as the replication tests are directional (i.e., the effect in the replication needs to be in the same direction as the original study) and two-sided tests are used, the false positive risk in each test is only 2.5% (so the total false positive risk with our two-stage procedure does still not exceed 5%; we ran a simulation to estimate the false positive risk more exactly with our two-stage procedure and the false positive risk is 4.2%). Related to this our power estimations, of the power to detect 50% of the original effect size in the first and second data collection pooled, is somewhat conservative as it does not take into account the dependency of the Stage 1 and Stage 2 tests. We ran a simulation to estimate the power more exactly of our two-stage testing procedure and the power to detect 50% of the original effect size is 91.1% instead of 90%. In some replications the statistical power was slightly larger than 90% as the total number of observations needed to be evenly divisible by some number (e.g., subjects or groups needed to be evenly divided into treatments). In some replications the sample sizes were also slightly larger than the planned sample size (as some original studies used exclusion criteria and the number of exclusions were not known in advance, it was difficult to collect exactly the planned number of observations).”

We now include this further explanation in our Methods section.

For secondary data, the sample size justification that was followed was: “the sample size required to achieve 50% power to detect 100% of the original effect size (minimum threshold sample size).” This is completely inadequate. Studies that have 50% power lead to no better rate of claims if there is a true effect than a coin flip – and a coin flip is a lot cheaper. Maybe this is a typo – because the authors did not share sufficient information about the studies (e.g., a link to the OSF) I can only take the analyses as briefly summarized in the article. The secondary replication studies are therefore not informative.

→ We have worked with a separate team to rerun all power calculations and we have updated Table 1 to include a column stating the power of the replications to detect 75%

of the original studies' effect size. The Table already includes a link to each replication's OSF repository in the "RR project" column, but we also added a detailed table of links to all relevant aspects of the studies' preregistrations in the Supplementary Materials (Table SM1). We have redone the analysis after filtering the replicated papers down to only those who's replication study achieved 80% power to detect 75% of the original effect size. This reduced our dataset of 35 replication studies to 28, including all new data replications, with which we can be justifiably confident in the replication outcome. This also necessitated the fit of slightly simplified models (one fewer random effect) to achieve satisfactory model fits with the smaller sample size. The updated model formulas are defined in Table M4.

For example, take the study by Simione & Gnagnarella, 2020. The replication yielded a $p = .214$. With 50% power, that is not unlikely even if there is a true effect. The replication of Seale et al., 2020 has a much smaller sample size as the original as well, and a non-significant result, as does Flesia et al., 2020. Messner & Payson, 2020 has a $p = 0.089$ in the replication.

→ Simione & Gnagnarella, 2020 and Messner & Payson 2020 were indeed underpowered and were excluded from our analysis. The power calculations for Seale et al 2020 and Flesia et al 2020 were revised following our checks. See Table 1

4. Secondary data collection should be separated from novel data collection.

In general the secondary data replications are therefore peculiar. Conceptually, they are also a bit of a different beast than replication studies based on new data. I do not believe both types of studies should be combined in a single analysis. First, their Type 2 error rate was less controlled. But more importantly, the predictions were not made based on the idea that these replications would only have 50% power, as the instructions of the participants, if I understand it well enough, was:

Note that both experts and novices were prompted to imagine that "all replication studies have high power (90% power to detect an effect 50-75% of the original effect size with $\alpha=0.05$, two-sided)".

As this is not true for the secondary data replications, why did the authors believe that these studies could be included in the analyses of the prediction success? I think moving the secondary data collections to a supplement is best, given how uninformative they are, and how they can not be used in computing prediction success, as the predictors were asked to imagine the claims being tested with higher power.

→ See comments above. We also note we now distinguish in multiple places throughout the paper between performance on new data and secondary data replications, including in Figure 2, Results/Section 4, and Discussion. We hope this provides enough information about the differences in performance on the two types of replications for readers to draw their own conclusions.

5. Checking all replication studies for deviations from the preregistered plan

The authors seem to be releasing all replication studies from this COVID project alongside this article. It took me a bit too long to figure out that Table 1 has the links to individual replication studies. I think at least one reviewer should spent time checking all these studies.

I started by checking one study, and found something that surprised me. I wanted to check study osf.io/kbm46 because the original had 78 countries, the replication had 103 countries. This seemed surprisingly low to me, given the sampling plan.

In the files <https://osf.io/cvmzh> I find the text:

For these calculations, the primary unit of analysis are Country. An estimate of the minimum viable sample size for the data analytic replication is: 41. For comparison, the stage1 required sample size would be: 201 and the stage2 sample size would be: 454. (Note, I want to give the author of this text credit for acknowledging “Confidence is low only because I am not deeply familiar with log linear.” for this power analysis.). Nevertheless, it seems like this study did not manage to collect the required data of 201 countries. The observed p-value was .14 in the replication. Was this a null due to low power?

I checked another study with a lower sample size in the replication than the original. osf.io/ytuk9 leads to <https://osf.io/926w4> which states:

For these calculations, the primary unit of analysis are participants. A preliminary estimate for the required sample size is ****a sample of 243 for stage 1 data collection**, collecting an additional 303 to reach ****a total of 546**** if stage 2 data collection is conducted.**

Despite a non-significant result after stage 1, with $p = .1203$, no second sample seems to have been collected, thus violating the sample size plan?

I wonder in how many studies there are unreported deviations from the plan described in the submitted manuscript, but these 2 checks make me believe someone should carefully check all studies.

→ *We include a section that describes all deviations from preregistrations in the supplemental information. These are classified according to varying degrees of deviations.*

If I read: <https://osf.io/5kcny> I see that the preregistration stated the number of novices as 45 and as 40 (so, the preregistration is not consistent). This sample size is not in line with the collected data – there were more novices, which is not a problem, but there is no explanation of why this is the case or how this happened, as the expectation was

“The number of novice participants is constrained by the availability of students in the relevant classes at UNC”.

→ *We now fully explain the deviations from our pre-registered plan in the Methods section. The inconsistency is regrettably a typo, which we also now explain in the Methods section.*

This link osf.io/t9xcn was not accessible, and should have been turned into a peer review only link before submission.

→ *This has now been fixed.*

6. Minor comment:

In the abstract the statement “Replication is an important “credibility control” mechanism” for clarifying the reliability of published findings.” is vague. A direct replication (as performed here) helps to identify Type 1 errors. Less direct replications help to test generalizability across the dimensions that are varied. Terms such as ‘credibility’ and ‘reliability’ are unnecessarily vague – they can be used, but then they need to be explicitly linked to epistemological goals. Credibility could be linked to an increase in subjective belief, and reliability to tests of validity (such as generalizability) – but such links and definitions need to be explicit.

→ *We thank the reviewer for helping us clarify this point. We no longer use ‘credibility’ and the new framing and discussion better explain how we use ‘reliability’.*

Use of qualifiers throughout the text. In the abstract, authors write “claims were strongly correlated ($r=.48$).” Why is this deemed a ‘strong’ correlation? It is a very low correlation if one would consider if the groups are equivalent (as there is clear variation between groups). What is it ‘strong’ for?

→ *We have now revised our interpretation of the strength of correlations, following Dancey C.P., Reidy J. Pearson Education; 2007. Statistics without Maths for Psychology.*

Symbols are not printed in the sentence: three categories (“ $_ \leq 0.001$ ”, “ $0.001 < _ \leq 0.01$ ”, “ $_ > 0.01$ ”),

→ *Now fixed.*

Round 2 ($\beta = 0547$, > a comma must be missing somewhere.

→ *Now fixed.*

Signed,

Daniel Lakens

FOLLOW-UP EDITOR'S COMMENTS

Following our decision letter, we had further interactions with Reviewers 1 and 3. As you are aware, the reviewers' statistical background differs (Bayesian vs Neyman-Pearson, respectively). Reviewer 3 expressed concerns that switching to a Bayesian analysis is not appropriate for the following two reasons:

- 1) Participants predicting success/failure to replicate were told that the replications were performed with 95% power and an alpha of 5%. The second is not true in one type of replication, the first not true in the secondary data analysis replications.
- 2) It is not possible to classify RRs as replicated vs non-replicated using Bayesian statistics, as Bayes Factors simply express the relative likelihood of 2 hypotheses. To make a 'replicated vs non-replicated' claim, a cut-off needs is needed and this cannot be done after the fact.

Reviewer 3 recommends instead to use an appropriate error correction approach applied to all sequential designs that sufficiently controls the error rate given the design that was used.

We shared this feedback with Reviewer 1, who disagreed. In response to point 1 above, they noted:

" it is indeed true that, for a subset of the studies, the power was much lower. The judgments of the people can, in my opinion, still be interpreted -- the authors have people's opinion of how well the results would replicate with high power, and this ought to be adjusted downwards for the subset of studies with low power. Alternatively, the judgments can be interpreted as measuring people's confidence that the effect exists (and can be detected under favorable circumstances)."

In response to point 2 above, they noted that a binary classification is not necessary -- you can use the relative likelihoods, which would be more informative.

As you can see, there is disagreement between our two experts and what is clear to us is that there are two alternative ways of approaching the analyses in your manuscript: Bayesian or frequentist. Either approach is valid and our recommendation to you (also made by Reviewer 1) was to provide two complementary analyses (Bayesian and frequentist), which will be both informative and function as robustness test.

Although our recommendation is to provide both sets of analysis, we would like to leave it up to you to decide which approach to pursue. Regardless, however, we would strongly encourage you to enlist a statistical expert (or experts if you perform complementary Bayesian and frequentist analyses) as a co-author in order to ensure the soundness of your analyses.

→ *We have performed a modified version of our original frequentist approach in this resubmission that should address the root cause of the concern of Reviewer 3 that led to this modelling discourse. The major concern of Reviewer 3 stemmed from how we presented the statistical power of the replication studies in the original submission, which has led to some doubts over the usefulness of the data. The “50% power to detect 50% of the original effect size” rule was used as part of the process to determine the minimum acceptable sample size for the replication studies but was not the statistical power of the completed replication studies themselves. In this resubmitted analysis we restricted our accuracy-based analyses to only include replication outcomes from studies that achieved $\geq 80\%$ power to detect 75% of the original effect size. This reduced the set of replication studies included in these analyses from 35 to 28 and retained 100% of the new data replications. We also agree with Reviewer 3 that a Bayesian framework would not have provided additional support to our analysis and have stayed with our original frequentist approach. By clarifying the actual statistical power of the replication studies and setting a minimum power threshold for inclusion in the analysis we believe we have addressed the core concern above.*

Decision Letter, first revision:

13th December 2023

Dear Dr Marcoci,

Thank you once again for your revised manuscript, entitled "Predicting the replicability of social and behavioural science claims in a crisis: The COVID-19 Preprint Replication Project," and for your patience during the re-review process.

Your manuscript has now been evaluated by the same reviewers who evaluated your original manuscript. All reviewer feedback is included at the end of this letter. Although the reviewers found your manuscript to have improved during revision, they also raise some important outstanding concerns. We remain interested in the possibility of publishing your study in Nature Human Behaviour, but would like to consider your response to these outstanding concerns in the form of a revised manuscript before we make a decision on publication.

Please address all remaining concerns of Reviewer 3. Specifically, we ask you to be fully explicit of all methodological shortcomings, as pointed out by the reviewer. While we appreciate the concerns that the reviewer raises about the power analysis. We ask you to be formally accurate and report power analyses for an alpha of 2.5% and revise the work in light of the resulting power estimates.

In sum, we invite you to revise your manuscript taking into account all reviewer and editor comments. We are committed to providing a fair and constructive peer-review process. Do not hesitate to contact us if there are specific requests from the reviewers that you believe are technically impossible or unlikely to yield a meaningful outcome.

We hope to receive your revised manuscript within 4-8 weeks. I would be grateful if you could contact us as soon as possible if you foresee difficulties with meeting this target resubmission date.

- Include a "Response to the editors and reviewers" document detailing, point-by-point, how you addressed each editor and referee comment. If no action was taken to address a point, you must provide a compelling argument. This response will be used by the editors and reviewers to evaluate your revision.
- Highlight all changes made to your manuscript or provide us with a version that tracks changes.

[REDACTED]

We look forward to seeing the revised manuscript and thank you for the opportunity to review your work. Please do not hesitate to contact me if you have any questions or would like to discuss these revisions further.

Sincerely,

[REDACTED]

REVIEWER COMMENTS:

Reviewer #1:

Remarks to the Author:

This is a thorough revision that adequately addresses my initial comments. I was positive about the original manuscript, and I am positive about this revision as well.

Minor comments

a. The abstract appears to contain some important typos (the same numbers are reported for the different conditions): "For experienced groups, average accuracy was 0.57 (95% CI: [0.52, 0.62]) after interaction and they correctly classified 60% of claims; beginners' average accuracy was 0.57 (95% CI: [0.52, 0.62]), correctly classifying 67% of claims."

b. It was disappointing to see that the authors spurned the Bayesian analysis, but I'll get over it.

I always sign my reviews,
E.J. Wagenmakers

Reviewer #2:

Remarks to the Author:

I think the authors have addressed mine and the other peer reviewers' comments satisfactorily, but I admit I have less statistical expertise than Reviewer 1 and 3, so defer to their judgement.

Reviewer #3:

Remarks to the Author:

It is interesting to see that based on my comments 7 studies are now no longer seen as worthy of being included in the analysis. On the one hand, I am of course happy to see my comments mattered. On the other hand, it worries me enormously that in such a large team of authors, and when peer reviewed by 2 other reviewers, no one noticed the points I observed, that led to a substantial change in the manuscript. I think we need to seriously reconsider how to review extremely large projects like this, as well as how projects like this guarantee internal quality control. There is so much material to look at and think about, I fear the current 3 reviewers are not enough. I consider this especially important because these large project get a lot of attention, and scrutiny. We have a duty to keep ourselves to the highest possible standards.

I think the rewrite where there are 'beginners' and 'experts' – none of which have domain knowledge, which is now clearly stated – resolves my earlier concerns about this topic. It is now communicated in a way that makes it relatively expected that there are no differences. The authors have not addressed my concern about explaining mechanisms through which people would make better predictions if they work a little bit longer in an unrelated scientific field, but they do acknowledge this study was not really a good test of this prediction, which I agree with. I also appreciate noting the lack of generalizability.

You failed to update the text in all relevant locations (e.g., figure label figure 2, some other places,

just search for the target words), so please double check this. I think we probably want to be very careful using the word 'expert' in our scientific field as a matter of justified humility, and an interesting paper on this is: Uygun Tunç, D. (2022). We should redefine scientific expertise: An extended virtue account. *European Journal for Philosophy of Science*, 12(4), 71. <https://doi.org/10.1007/s13194-022-00498-2>

I think the criticism I had on the fact that sequential analyses are performed, but that this was not acknowledged, is only partly dealt with. The authors correctly chose to stick with their frequentist analysis, but some points remain. First, the authors write "Note that one could object that we are informally performing sequential analyses using a 5% alpha level at each look at the data and thus inflating the Type 1 error rate". I would like to see this changed in the statement of 1) an explicit acknowledgement that the authors performed sequential analyses informally, and that they should have incorporated this into their design if they wanted to follow best practices. This is not contentious, and unless we want future replication projects to make the same mistakes, we need to call a stop to this bad research practice here and now. The authors should note that future studies should correctly design sequential studies if they use the same multi-stage design. Then 2) the authors correctly note they are saved by their incoherent analysis plan. It is correct that their error rate is in the end not inflated above 5% as they factually performed a one-sided test at 2.5%. Nevertheless, this is also a practice that should not be emulated, and authors should have performed one-sided tests at 5% because the 2-sided test can not answer the question they asked. As explained by Kaiser, 1960, it is logically incoherent to make a directional claim after a two-sided test. So, the authors again need to acknowledge they performed two-sided tests but that future studies should not make the same mistake. Finally, 3) the authors then have to admit this leads to an incoherent design, as they did not power their studies for an alpha level of 0.025%, but 5%. The statement that "Power estimates are somewhat conservative" can not be made without an extensive investigation of the consequences of inflated power estimates (as they are now computed based on an alpha of 5%). What to do about this mess? At the very least, the authors should probably recommend future studies enlist the help of a methodologist to prevent these mistakes. As Fisher already noted: "To call in the statistician after the experiment is done may be no more than asking him to perform a post-mortem examination". I assume the power reported in Table 1 (which is a nice addition) is computed for an alpha of 5%, not an alpha of 2.5%, or even 4.2%? If the authors choose 2.5%, I fear some additional studies might drop below 80% power, and the entire result section needs to be rewritten and the sample size is a bit smaller. That would make the paper more formally accurate but it is also a lot of work. Alternatively, the authors can not that the sequential design leads to some issues with the alpha level, not that the true alpha level is somewhere between 0.025 and 0.5, that the power values are not completely accurate, and give the recommendation that in the future these issues are addressed before data collection.

I looked at several raw files in each preregistration. Almost in every file important uncertainties are noted by the researchers creating these files. This information is important, and I believe it would be reported in most publications if the authors would write a single paper on each study. For example, from the earlier review the sentence "Confidence is low only because I am not deeply familiar with log linear" is quite a substantial comment about how a power analysis is done. All these aspects of studies are completely ignored in this review, but if this study was submitted as a single paper, the power analysis might lead to rejection of the study, with the suggestion to redo the power analysis and compute the appropriate sample (and of course this single sentence is just one example – this power analysis might actually be perfect). I feel that is not fair, not smart. There are no standards how to acknowledge such facts. But some general statement that in each project, there can be more or less

substantive issues that emerged while designing the studies, which are probably open to criticism and sometimes deserve to be examined in more detail, were not checked or peer reviewed. Therefore, there is a substantial probability that anyone going through each replication study will identify at least some errors that throw decent doubt on a single study – and maybe on several studies. I do not think it can be avoided with a project of this size, in this timeframe, but I feel this should be acknowledged. Each individual study has not been scrutinized as much as normal single studies, and this uncertainty needs to be acknowledged, I feel. I think most meta-scientists will disagree with me, but I think we are going to get this criticism in the future, and it is better to pre-empt it. It might also be good to say where you will correct mistakes that are identified in the future (for example on the OSF page) as it seems inevitable that some mistakes in the individual studies were not communicated in the overall paper.

Minor:

You tried to improve your writing, by changing the statement that correlations were weak, to saying they were not notable. 'Not notable' is even worse than 'weak'. If you can interpret the effect size, please do so. If you can not, just remove all qualifiers. Throughout, effect sizes are just reported, but never meaningfully interpreted. I understand this is difficult, but if you can't, adding nonsense words does not make things better.

The authors write "We have worked with a separate team to rerun all power calculations". Where are the analysis scripts for this? And who was this team, as the authors do not seem to have changed. They deserve to be named, and acknowledged, at least? Did they have sufficient experience to be in the 'experienced' group? (The last sentence is a joke).

Signed,

Daniel Lakens

Author Rebuttal, first revision:

Reviewer #1:

Remarks to the Author:

This is a thorough revision that adequately addresses my initial comments. I was positive about the original manuscript, and I am positive about this revision as well.

→ *We greatly appreciate all the work reviewer 1 has put into commenting on our manuscript.*

Minor comments

a. The abstract appears to contain some important typos (the same numbers are reported for the different conditions): "For experienced groups, average accuracy was 0.57 (95% CI: [0.52, 0.62]) after interaction and they correctly classified 60% of claims; beginners' average accuracy was 0.57 (95% CI: [0.52, 0.62]), correctly classifying 67% of claims."

→ *We thank reviewer 1 for noticing this error. We have now made the relevant changes.*

b. It was disappointing to see that the authors spurned the Bayesian analysis, but I'll get over it.

→ *We are grateful to reviewer 1 for their suggestions regarding a Bayesian analysis and although we didn't take this path here, we acknowledge it would be interesting to explore it in future work.*

I always sign my reviews,
E.J. Wagenmakers

Reviewer #2:

Remarks to the Author:

I think the authors have addressed mine and the other peer reviewers' comments satisfactorily, but I admit I have less statistical expertise than Reviewer 1 and 3, so defer to their judgement.

→ *We greatly appreciate all the work reviewer 2 has put into commenting on our manuscript.*

Reviewer #3:

Remarks to the Author:

It is interesting to see that based on my comments 7 studies are now no longer seen as worthy of being included in the analysis. On the one hand, I am of course happy to see my comments mattered. On the other hand, it worries me enormously that in such a large team of authors, and when peer reviewed by 2 other reviewers, no one noticed the points I observed, that led to a substantial change in the manuscript. I think we need to seriously reconsider how to review extremely large projects like this, as well as how projects like this guarantee internal quality control. There is so much material to look at and think about, I fear the current 3 reviewers are not enough. I consider this especially important because these large project get a lot of attention, and scrutiny. We have a duty to keep ourselves to the highest possible standards.

→ *We are very grateful to Reviewer 3 for their scrutiny and the attention they paid to our work. We believe guidelines and best practices for ensuring internal quality control of large projects such as ours would be a very welcome contribution to community-led initiatives. To increase the reliability of our results, we adopted internal quality control protocols as detailed in Methods, we shared all our underlying datasets, and we now report all results included in the Main Text both for the entire corpus and for only those research claims that achieved enough power using an $\alpha = 2.5\%$ in the supplementary materials to allow readers to make their own decisions about the*

support our data provide for our conclusions. We believe in such large studies transparency is essential and we welcomed the reviewer's suggestions about how to more transparently share our data so that others can continue to scrutinize our work post-publication.

I think the rewrite where there are 'beginners' and 'experts' – none of which have domain knowledge, which is now clearly stated – resolves my earlier concerns about this topic. It is now communicated in a way that makes it relatively expected that there are no differences. The authors have not addressed my concern about explaining mechanisms through which people would make better predictions if they work a little bit longer in an unrelated scientific field, but they do acknowledge this study was not really a good test of this prediction, which I agree with. I also appreciate noting the lack of generalizability.

→ We thank the reviewer for pushing us on this point and we believe it has made our manuscript clearer. We do not share their intuition regarding the expected difference in performance between our two groups, but we are content that in the way it's written readers can make their own minds about the value of the findings reported. The reviewer is right that we did not include a discussion about the presumed causal mechanisms, however, given how we are now framing the study we thought such a discussion was no longer necessary (and we are already struggling to contain the main text of the manuscript within the word count). We thank the reviewer for understanding.

You failed to update the text in all relevant locations (e.g., figure label figure 2, some other places, just search for the target words), so please double check this. I think we probably want to be very careful using the word 'expert' in our scientific field as a matter of justified humility, and an interesting paper on this is: Uygun Tunç, D. (2022). We should redefine scientific expertise: An extended virtue account. *European Journal for Philosophy of Science*, 12(4), 71. <https://doi.org/10.1007/s13194-022-00498-2>

→ We thank the reviewer for noticing these lapses and we have now updated the text throughout.

I think the criticism I had on the fact that sequential analyses are performed, but that this was not acknowledged, is only partly dealt with. The authors correctly chose to stick with their frequentist analysis, but some points remain. First, the authors write "Note that one could object that we are informally performing sequential analyses using a 5% alpha level at each look at the data and thus inflating the Type 1 error rate". I would like to see this changed in the statement of 1) an explicit acknowledgement that the authors performed sequential analyses informally, and that they should have incorporated this into their design if they wanted to follow best practices. This is not contentious, and unless we want future replication projects to make the same mistakes, we need to call a stop to this bad research practice here and now. The authors should note that future studies should correctly design sequential studies if they use the same multi-stage design.

→ *We now make an explicit statement about performing sequential analyses informally. We also include a statement in the discussion (main text) about how the design of future studies could be improved by adopting best practices and using sequential analyses.*

Then 2) the authors correctly note they are saved by their incoherent analysis plan. It is correct that their error rate is in the end not inflated above 5% as they factually performed a one-sided test at 2.5%. Nevertheless, this is also a practice that should not be emulated, and authors should have performed one-sided tests at 5% because the 2-sided test can not answer the question they asked. As explained by Kaiser, 1960, it is logically incoherent to make a directional claim after a two-sided test. So, the authors again need to acknowledge they performed two-sided tests but that future studies should not make the same mistake.

→ *We now make this recommendation explicitly in the discussion (main text).*

Finally, 3) the authors then have to admit this leads to an incoherent design, as they did not power their studies for an alpha level of 0.025%, but 5%. The statement that “Power estimates are somewhat conservative” can not be made without an extensive investigation of the consequences of inflated power estimates (as they are now computed based on an alpha of 5%).

→ *We have now recalculated our power estimates based on an $\alpha = 2.5\%$ and present results for both α -levels side-by-side in the paper (main text) as well as in a new Appendix (Supplementary Materials 2).*

What to do about this mess? At the very least, the authors should probably recommend future studies enlist the help of a methodologist to prevent these mistakes. As Fisher already noted: “To call in the statistician after the experiment is done may be no more than asking him to perform a post-mortem examination”.

I assume the power reported in Table 1 (which is a nice addition) is computed for an alpha of 5%, not an alpha of 2.5%, or even 4.2%? If the authors choose 2.5%, I fear some additional studies might drop below 80% power, and the entire result section needs to be rewritten and the sample size is a bit smaller. That would make the paper more formally accurate but it is also a lot of work.

Alternatively, the authors can not that the sequential design leads to some issues with the alpha level, not that the true alpha level is somewhere between 0.025 and 0.5, that the power values are not completely accurate, and give the recommendation that in the future these issues are addressed before data collection.

→ *We have now revised Table 1 to also include power calculations for $\alpha = 2.5\%$ and included a discussion about the different design options faced by large-scale replication studies such as ours, including a table of comparisons between our results conditional on different choices of α . Finally,*

we added a new Appendix (Supplementary Materials 3) where we present all results reported in the Main Text for claims that reached the power threshold using an $\alpha = 2.5\%$.

I looked at several raw files in each preregistration. Almost in every file important uncertainties are noted by the researchers creating these files. This information is important, and I believe it would be reported in most publications if the authors would write a single paper on each study. For example, from the earlier review the sentence “Confidence is low only because I am not deeply familiar with log linear” is quite a substantial comment about how a power analysis is done. All these aspects of studies are completely ignored in this review, but if this study was submitted as a single paper, the power analysis might lead to rejection of the study, with the suggestion to redo the power analysis and compute the appropriate sample (and of course this single sentence is just one example – this power analysis might actually be perfect). I feel that is not fair, not smart. There are no standards how to acknowledge such facts. But some general statement that in each project, there can be more or less substantive issues that emerged while designing the studies, which are probably open to criticism and sometimes deserve to be examined in more detail, were not checked or peer reviewed. Therefore, there is a substantial probability that anyone going through each replication study will identify at least some errors that throw decent doubt on a single study – and maybe on several studies. I do not think it can be avoided with a project of this size, in this timeframe, but I feel this should be acknowledged. Each individual study has not been scrutinized as much as normal single studies, and this uncertainty needs to be acknowledged, I feel. I think most meta-scientists will disagree with me, but I think we are going to get this criticism in the future, and it is better to pre-empt it. It might also be good to say where you will correct mistakes that are identified in the future (for example on the OSF page) as it seems inevitable that some mistakes in the individual studies were not communicated in the overall paper.

→ Thank you very much for your comment. We have included an additional statement in the methods section acknowledging this limitation, encouraging review of content on OSF, and explaining how the project contents will be maintained and updated for the individual replication studies similar to other large-scale replication projects.

Minor:

You tried to improve your writing, by changing the statement that correlations were weak, to saying they were not notable. ‘Not notable’ is even worse than ‘weak’. If you can interpret the effect size, please do so. If you can not, just remove all qualifiers. Throughout, effect sizes are just reported, but never meaningfully interpreted. I understand this is difficult, but if you can’t, adding nonsense words does not make things better.

→ Thank you for your comment. We reviewed our statements on effect sizes and adapted the phrasings. For instance, we removed the description “not notable” throughout.

The authors write “We have worked with a separate team to rerun all power calculations”. Where are the analysis scripts for this? And who was this team, as the authors do not seem to have changed. They deserve to be named, and acknowledged, at least? Did they have sufficient experience to be in the ‘experienced’ group? (The last sentence is a joke).

→ *In the acknowledgement we include "...thank Marcel van Assen, Robbie van Aert, Marjan Bakker, and Maksim Sitnikov for help with power calculations and calculation of effect sizes...". This was their preference. The scripts are linked to the OSF project containing the replication outcomes (<https://osf.io/fjksb/>).*

Signed,

Daniel Lakens

Decision Letter, second revision:

30th May 2024

Dear Dr. Marcoci,

Thank you for your patience as we've prepared the guidelines for final submission of your Nature Human Behaviour manuscript, "Predicting the replicability of social and behavioural science claims in a crisis: The COVID-19 Preprint Replication Project" (NATHUMBEHAV-23020399B). Please carefully follow the step-by-step instructions provided in the attached file, and add a response in each row of the table to indicate the changes that you have made. Please also check and comment on any additional marked-up edits we have proposed within the text. Ensuring that each point is addressed will help to ensure that your revised manuscript can be swiftly handed over to our production team.

We would hope to receive your revised paper, with all of the requested files and forms within two-three weeks. Please get in contact with us if you anticipate delays.

Nature Human Behaviour offers a Transparent Peer Review option for new original research manuscripts submitted after December 1st, 2019. As part of this initiative, we encourage our authors to support increased transparency into the peer review process by agreeing to have the reviewer comments, author rebuttal letters, and editorial decision letters published as a Supplementary item.

When you submit your final files please clearly state in your cover letter whether or not you would like to participate in this initiative. Please note that failure to state your preference will result in delays in accepting your manuscript for publication.

In recognition of the time and expertise our reviewers provide to Nature Human Behaviour's editorial process, we would like to formally acknowledge their contribution to the external peer review of your manuscript entitled "Predicting the replicability of social and behavioural science claims in a crisis: The COVID-19 Preprint Replication Project". For those reviewers who give their assent, we will be publishing their names alongside the published article.

Cover suggestions

We welcome submissions of artwork for consideration for our cover. For more information, please see our guide for cover artwork.

ORCID

Non-corresponding authors do not have to link their ORCIDs but are encouraged to do so. Please note that it will not be possible to add/modify ORCIDs at proof. Thus, please let your co-authors know that if they wish to have their ORCID added to the paper they must follow the procedure described in the following link prior to acceptance: <https://www.springernature.com/gp/researchers/orcid/orcid-for-nature-research>

Nature Human Behaviour has now transitioned to a unified Rights Collection system which will allow our Author Services team to quickly and easily collect the rights and permissions required to publish your work. Approximately 10 days after your paper is formally accepted, you will receive an email in providing you with a link to complete the grant of rights. If your paper is eligible for Open Access, our Author Services team will also be in touch regarding any additional information that may be required to arrange payment for your article.

Please note that *Nature Human Behaviour* is a Transformative Journal (TJ). Authors may publish their research with us through the traditional subscription access route or make their paper immediately open access through payment of an article-processing charge (APC). Authors will not be required to make a final decision about access to their article until it has been accepted. Find out more about Transformative Journals

Authors may need to take specific actions to achieve compliance with funder and institutional open access mandates. If your research is supported by a funder that requires immediate open access (e.g. according to Plan S principles) then you should select the gold OA route, and we will direct you to the compliant route where possible. For authors selecting the subscription publication route, the journal's standard licensing terms will need to be accepted, including self-

archiving policies. Those licensing terms will supersede any other terms that the author or any third party may assert apply to any version of the manuscript.

[REDACTED]

Best regards,
[REDACTED]

On behalf of

[REDACTED]

Reviewer #3:

Remarks to the Author:

The reviewers have incorporated the final suggestions I have made. I have no further points that should be addressed. I appreciate the authors responsiveness to my comments, and sincerely hope they feel my comments on the paper helped prevent some criticism after publication.

Final Decision Letter:

Dear Dr Marcoci,

We are pleased to inform you that your Article "Predicting the replicability of social and behavioural science claims in COVID-19 preprints", has now been accepted for publication in *Nature Human Behaviour*.

Please note that *Nature Human Behaviour* is a Transformative Journal (TJ). Authors may publish their research with us through the traditional subscription access route or make their paper immediately open access through payment of an article-processing charge (APC). Authors will not be required to make a final decision about access to their article until it has been accepted. Find out more about Transformative Journals

Authors may need to take specific actions to achieve compliance with funder and institutional open access mandates. If your research is supported by a funder that requires immediate open access (e.g.

according to Plan S principles) then you should select the gold OA route, and we will direct you to the compliant route where possible. For authors selecting the subscription publication route, the journal's standard licensing terms will need to be accepted, including self-archiving policies. Those licensing terms will supersede any other terms that the author or any third party may assert apply to any version of the manuscript.

Once your manuscript is typeset and you have completed the appropriate grant of rights, you will receive a link to your electronic proof via email with a request to make any corrections within 48 hours. If, when you receive your proof, you cannot meet this deadline, please inform us at risproduction@springernature.com immediately. Once your paper has been scheduled for online publication, the Nature press office will be in touch to confirm the details.

To assist our authors in disseminating their research to the broader community, our SharedIt initiative

provides you with a unique shareable link that will allow anyone (with or without a subscription) to read the published article. Recipients of the link with a subscription will also be able to download and print the PDF.

With best regards,
[REDACTED]